# Faithful Group Shapley Value

**Kiljae Lee**[*]
The Ohio State University
lee.10428@osu.edu

**Ziqi Liu**[*]
Carnegie Mellon University
ziqiliu2@andrew.cmu.edu

**Weijing Tang**
Carnegie Mellon University
weijingt@andrew.cmu.edu

**Yuan Zhang**[†]
The Ohio State University
yzhanghf@stat.osu.edu

## Abstract

Data Shapley is an important tool for data valuation, which quantifies the contribution of individual data points to machine learning models. In practice, group-level data valuation is desirable when data providers contribute data in batch. However, we identify that existing group-level extensions of Data Shapley are vulnerable to *shell company attacks*, where strategic group splitting can unfairly inflate valuations. We propose Faithful Group Shapley Value (FGSV) that uniquely defends against such attacks. Building on original mathematical insights, we develop a provably fast and accurate approximation algorithm for computing FGSV. Empirical experiments demonstrate that our algorithm significantly outperforms state-of-the-art methods in computational efficiency and approximation accuracy, while ensuring faithful group-level valuation.

## 1 Introduction

As data become increasingly crucial in modern machine learning, quantifying its value has significant implications for faithful compensation and data market design [29, 39]. The Shapley value, a foundational concept from cooperative game theory [30], stands out as the unique valuation method satisfying four desirable axioms for faithful data valuation [7, 14]. Consequently, it has been applied to diverse applications, including collaborative intelligence [22, 25] and copyright compensation [35], and quantifying feature importance in explainable AI [26].

Many real-world scenarios demand evaluation of *data sets* instead of individual data points. In applications like data marketplaces [39] or compensation allocation for generative AI [4, 35], data are contributed by owners who possess entire datasets, making group-level data valuation a natural choice. A more technical motivation is that individual-level Shapley values are often computationally challenging to approximate for big data; while group-level valuation is much faster and can provide useful insights [17]. In explainable AI, group-level feature importance reportedly provides more robust and interpretable valuation [32, 3].

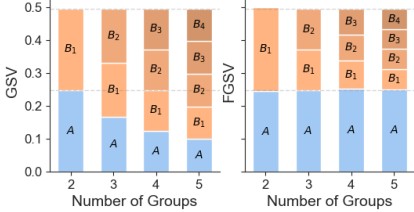

Figure 1: Left: GSV; right: FGSV (our method). Vertical span: valuation. Group $A$ is fixed; group $B$ engages increasing degrees of *shell company attack* (left→right). Detailed experimental set-up in Appendix B.

---

[*]Lee and Liu equally contributed; they are co-first authors and were listed alphabetically.

[†]Corresponding author.

As group data valuation frameworks are increasingly used in real-world scenarios such as data markets and copyright compensation, their robustness against adversarial manipulation becomes important. Prior studies on attacks and defenses in data valuation have focused primarily on the *individual-level* data valuation, such as the *copier attack* that duplicates existing data to unjustly inflate value [6, 10]. In contrast, we identify a new vulnerability unique to group data valuation, which has been largely unexplored in literature. Existing extensions of the Shapley framework to group data valuation treat each pre-defined group as an atomic unit and then apply the standard Shapley value formulation [15, 20, 35, 32], which we refer to as Group Shapley Value (GSV). However, we demonstrate—both theoretically and empirically—that this approach suffers from susceptibility to strategic manipulation through partitioning. Consider the valuation of a fixed data group $A$. As shown in Figure 1, partitioning the remaining data points into smaller subgroups reduces the GSV for $A$, despite no change in data content. In other words, malicious players may exploit the loophole in GSV by splitting their data among puppy subsidiaries, a vulnerability we coin as **shell company attack**.

To address this issue, we propose a faithfulness axiom for group data valuation, such that the total valuation of the same set of data remains unchanged, regardless of how others are subdivided. Based on this, we introduce the **Faithful Group Shapley Value (FGSV)** for a group as the sum of individual Data Shapley values of its members, and prove that it uniquely satisfies the set of axioms that are desirable for faithful group data valuation. Figure 1 demonstrates that FGSV effectively defends against the shell company attack.

While FGSV safeguards faithfulness, its exact computation requires combinatorial computation of individual Shapley values. Although numerous approximation algorithms for individual Shapley values have been developed [2, 7, 14, 34, 26, 3, 38, 24, 27], simply applying them and summing up results in each group is slow for large groups and may compound approximation errors. Importantly, we proved that a small subset of terms dominate in FGSV's formula. As a result, we developed an original algorithm that directly approximates FGSV fast and accurately. Our numerical experiments demonstrate its computational efficiency and approximation accuracy against summing up individual Shapley values computed by state-of-the-art (SOTA) methods. We also applied FGSV to faithful copyright compensation on Stable Diffusion models for image generation.

## 2 Preliminaries: individual and group Shapley values

**Individual Shapley value (SV).** Let $\mathcal{D} = \{z_1, \ldots, z_n\} \in \mathcal{Z}^n$ denote the training data, where $n := |\mathcal{D}|$. Let $S \subseteq [n]$ be an index set of cardinality $s := |S|$, and write $\mathcal{S} := \{z_i : i \in S\}$. A *utility function* $U(\mathcal{S})$ assigns a performance score (such as classification accuracy) to method trained based on data $\mathcal{S}$. To simplify notation, we may refer to "$U(\mathcal{S})$" by index as "$U(S)$" and write them interchangeably. The *individual Shapley value* of data point $i$ is defined as

$$\text{SV}(i) := \sum_{S \subseteq [n] \setminus \{i\}} \frac{|S|!(n - |S| - 1)!}{n!} \{U(S \cup \{i\}) - U(S)\}. \tag{1}$$

The widespread adoption of Shapley value stems from its strong theoretical foundation that it uniquely satisfies four desired axioms, called *null player*, *symmetry*, *linearity*, and *efficiency*. We will review these principles as part of our axiomatization for group Shapley value in Definition 2.

**Group Shapley value (GSV).** As mentioned in Section 2, most group-level Shapley values [15, 35, 32] adopt the so-called *group-as-individual* (GaI) approach. Suppose the entire data set is partitioned into $(K + 1)$ disjoint groups: $\mathcal{D} = S_0 \cup S_1 \cup \cdots \cup S_K$, in which, $S_0 \subseteq [n]$ is the group we aim to evaluate. The *group Shapley value (GSV)* of $S_0$, as in [15, 35, 32], is defined as

$$\text{GSV}(S_0) = \sum_{I \subseteq \{1, \ldots, K\}} \frac{|I|!(K - |I|)!}{(K + 1)!} \left\{ U\left(\left\{\cup_{k \in I} S_k\right\} \cup S_0\right) - U\left(\cup_{k \in I} S_k\right) \right\}. \tag{2}$$

Clearly, (2) is a "set version" of (1). Thus GSV also satisfies the four axioms for SV at the group level. However, it is questionable whether these axioms, originally formulated for individual-level valuation, are appropriate for groups.

**Fairness issue with GSV.** The GSV valuation of $S_0$ as in (2) can be impacted by how the rest of the data are grouped. A serious consequence is that GSV is prone to the **shell company attack**: splitting a group into smaller subgroups will increase the total valuation earned by the same set of data points. We have seen an illustration in Figure 1 in Section 1. Now we theoretically formalize this observation.

**Proposition 1** (Shell company attack). *Let $\mathcal{P}$ be the underlying data distribution over the sample space $\mathcal{Z}$, and write $\mathcal{P}^s$ for the $s$-fold product distribution; thus $S \sim \mathcal{P}^s$ denotes an i.i.d. sample $S = (z_1, \ldots, z_s) \in \mathcal{Z}^s$ with $z_i \sim \mathcal{P}$. Let $\bar{U}(s) = \mathbb{E}_{S \sim \mathcal{P}^s}[U(S)]$ denote the expected utility for data $S$ (which only depends on $s = |S|$). Now we split a group $S_k$ into two non-empty subgroups $S_k'$ and $S_k''$ (i.e., $S_k' \cup S_k'' = S_k$, $S_k' \cap S_k'' = \emptyset$, and $S_k', S_k'' \neq \emptyset$), If $\bar{U}(s)$ satisfies a* prudence *condition:*

$$\Delta_s^3 \bar{U}(s) := \bar{U}(s+3) - 3\bar{U}(s+2) + 3\bar{U}(s+1) - \bar{U}(s) > 0, \tag{3}$$

*then*

$$\mathbb{E}[\mathrm{GSV}(S_k)] < \mathbb{E}[\mathrm{GSV}(S_k')] + \mathbb{E}[\mathrm{GSV}(S_k'')]. \tag{4}$$

Condition (3) is a familiar concept in economics, characterizing risk-aversion behaviors under uncertainty [8]. It is also observed in machine learning, where the performance of a learning method saturates as the same type of data repeatedly come in [18, 12].

## 3 Our method

### 3.1 Faithfulness axiom and Faithful Group Shapley Value (FGSV)

Motivated by the faithfulness issue, we introduce a set of axioms that formalize desirable principles for faithful group data valuation. We first axiomatize group-level data valuation that generalizes the Shapley's four axioms for individual valuation.

**Definition 1** (Group data valuation). *For data $\mathcal{D} = \{z_1, \ldots, z_n\}$, utility $U$ and partition $\Pi = \{S_1, \ldots, S_K\}$ such that $S_{k_1} \cap S_{k_2} = \emptyset, \forall 1 \leq k_1 < k_2 \leq n$ and $S_1 \cup \cdots \cup S_K = [n]$, a* group data valuation *method $\nu_{U,\mathcal{D},\Pi}(\cdot) : \Pi \to \mathbb{R}$ is a mapping that assigns a real-valued score to each $S_k \in \Pi$.*

Next, we propose a set of axioms that a faithful group data valuation method should satisfy.

**Definition 2** (Axioms for faithful group data valuation). *A group data valuation $\nu_{U,\mathcal{D},\Pi}$ is called* **faithful** *if it satisfies the following axioms for any dataset $\mathcal{D}$, utility $U$, and partition $\Pi$:*

1. **Null player**: *For any $S \in \Pi$, if every subset $S' \subseteq S$ satisfies $U(S'' \cup S') = U(S'')$ for all $S'' \subseteq [n] \setminus S'$, then $\nu_{U,\mathcal{D},\Pi}(S) = 0$.*

2. **Symmetry**: *For any $S_1, S_2 \in \Pi$, $|S_1| = |S_2|$, if there is a bijection $\sigma : S_1 \to S_2$, s.t. $U(S'' \cup S') = U(S'' \cup \sigma(S'))$ for all $S' \subseteq S_1$ and $S'' \subseteq [n] \setminus (S' \cup \sigma(S'))$, then $\nu_{U,\mathcal{D},\Pi}(S_1) = \nu_{U,\mathcal{D},\Pi}(S_2)$.*

3. **Linearity**: *For any utility functions $U_1, U_2$ and scalars $\alpha_1, \alpha_2 \in \mathbb{R}$, we have for all $S \in \Pi$ $\nu_{\alpha_1 U_1 + \alpha_2 U_2, \mathcal{D}, \Pi}(S) = \alpha_1 \nu_{U_1, \mathcal{D}, \Pi}(S) + \alpha_2 \nu_{U_2, \mathcal{D}, \Pi}(S)$.*

4. **Efficiency**: *For any partition $\Pi = \{S_1, \ldots, S_K\}$, we have $\sum_{k=1}^K \nu_{U,\mathcal{D},\Pi}(S_k) = U([n])$.*

5. **Faithfulness**: *For any group $S \in \Pi_1 \cap \Pi_2$, we have $\nu_{U,\mathcal{D},\Pi_1}(S) = \nu_{U,\mathcal{D},\Pi_2}(S)$.*

Notably, our *null player* and *symmetry* require group valuation to faithfully reflect the contributions of its individual members. They are strictly weaker assumptions than their GaI counterparts. For example, a group assigned zero value in GaI can receive nonzero value under our axioms if some of its members have marginal contributions, but not the other way around. *Linearity* and *efficiency* are standard group-level extensions. Our newly introduced Axiom 5 requires that a group's value is determined only on its own competitive merit and rules out the unfair competition means of shell company attack.

It turns out that there is a unique group data valuation method that can satisfy all axioms in Definition 2: simply add up all member's individual Shapley values.

**Theorem 1.** *The only group data valuation method $\nu_{U,\mathcal{D},\Pi}$ that satisfies all axioms in Definition 2 is $\nu_{U,\mathcal{D},\Pi}(S) = \sum_{i \in S} \mathrm{SV}(i)$, where $\mathrm{SV}(i)$ is the individual Shapley value defined in (1).*

In view of Theorem 1, we propose **Faithful Group Shapley Value (FGSV)**:

$$\mathrm{FGSV}(S_0) := \sum_{i \in S_0} \mathrm{SV}(i). \tag{5}$$

### 3.2 Fast and accurate approximation algorithm for FGSV

The exact evaluation of FGSV requires combinatorial computation. To develop a feasible approximation method, we make a series of key mathematical observations leading to an efficient algorithm.

To start, from the definition of FGSV (5) and (1), we see that $\mathrm{FGSV}(S_0)$ is a complicated linear combination of $U(S)$ terms, where $S$ ranges over all subsets of $[n]$. Therefore, the first step towards simplification is to discover the pattern in the coefficient in front of each $U(S)$ term.

**Key observation 1.** *In* $\mathrm{FGSV}(S_0)$, *the coefficient of* $U(S)$ *depends on* $S$ *and* $S_0$ *only through the tuple* $(s_1, s, s_0)$, *where recall that* $s_0 := |S_0|$ *and* $s := |S|$, *and define* $s_1 := |S_0 \cap S|$.

In other words, any two terms $U(S)$ and $U(S')$ with $|S| = |S'|$ and $|S_0 \cap S| = |S_0 \cap S'|$ share the same coefficient in $\mathrm{FGSV}(S_0)$. This motivates us to aggregate these terms in our analysis. Let $\mathscr{A}_{s,s_1} := \{S : |S| = s, |S \cap S_0| = s_1\}$ collect all $S$'es with the same $(s, s_1)$ configuration, and define

$$\mu\left(\frac{s_1}{s}; s, s_0, n\right) := \frac{\sum_{S \in \mathscr{A}_{s,s_1}} U(S)}{|\mathscr{A}_{s,s_1}|} = \frac{\sum_{S:|S|=s,|S\cap S_0|=s_1} U(S)}{\binom{s_0}{s_1}\binom{n-s_0}{s-s_1}}. \tag{6}$$

When sampling a subset $S \subseteq [n]$ of size $s$ without replacement, the size of $S \cap S_0$, which we now denote as the boldfaced $\boldsymbol{s_1}$ to emphasize its randomness, follows a hypergeometric distribution: $\mathbb{P}(\boldsymbol{s_1} = s_1) = \binom{s_0}{s_1}\binom{n-s_0}{s-s_1}/\binom{n}{s}$. Using this fact, we can re-express FGSV in terms of $\mu$.

**Lemma 1.** *Let* $\boldsymbol{s_1} \sim \mathcal{HG}(n, s_0, s)$. *We can rewrite FGSV as*

$$\mathrm{FGSV}(S_0) = \frac{s_0}{n}\left[U([n]) - U(\varnothing)\right] + \sum_{s=1}^{n-1} \mathcal{T}(s), \tag{7}$$

*where*

$$\mathcal{T}(s) := \mathbb{E}_{\boldsymbol{s_1} \sim \mathcal{HG}(n,s_0,s)}\left[\frac{n}{n-s}\left(\frac{\boldsymbol{s_1}}{s} - \frac{s_0}{n}\right)\mu\left(\frac{\boldsymbol{s_1}}{s}; s, s_0, n\right)\right]. \tag{8}$$

Instead of directly estimating $\mathcal{T}(s)$ via Monte Carlo, we discover two key observations that deepen our understandings of (7) and (8), building on which, we can greatly reduce computational cost.

**Key observation 2.** *The probability* $\mathbb{P}(\boldsymbol{s_1} = s_1)$ *decays exponentially in* $|s_1 - ss_0/n|$.

Key observation 2 implies that $\mathcal{T}(s)$ is dominated by the values of $\boldsymbol{s_1}$ around its mean $\mathbb{E}[\boldsymbol{s_1}] = ss_0/n$. To deepen our understanding of $\mathcal{T}(s)$, for this moment, we informally deem $\mu$ as a smooth function of the continuous variable $s_1/s$, with formal characterization provided later. Then, applying a "Taylor expansion" of $\mu(s_1/s; s, s_0, n)$ around $s_1/s = s_0/n$ leads to the following intuition.

**Key observation 3 (Informal).** $\mathcal{T}(s) \approx s^{-1}(s_0/n)(1 - s_0/n)\mu'(s_0/n; s, s_0, n)$.

Key observation 3 reveals that, under suitable conditions, the term $\mathcal{T}(s)$ can be efficiently estimated by evaluating the derivative $\mu'(\cdot; s, s_0, n)$ at a single point $s_0/n$.

Next, we formalize the above discoveries into rigorous mathematical results.

**Assumption 1** (Boundedness)**.** *For all* $s \in \mathbb{N}$ *and* $\mathcal{S} \in \mathcal{Z}^s$, $|U(\mathcal{S})| \leq C$ *for a universal constant* $C$.

**Assumption 2** (Second-order algorithmic stability of utility)**.** *There exist constants* $C > 0$ *and* $\upsilon > 0$ *such that for all* $s \in \mathbb{N}$, $\mathcal{S} \in \mathcal{Z}^s$ *and* $z_1, z_1', z_2, z_2' \in \mathcal{Z}$,

$$\left|U(\mathcal{S} \cup \{z_1, z_1'\}) - U(\mathcal{S} \cup \{z_1, z_2\}) - U(\mathcal{S} \cup \{z_1', z_2'\}) + U(\mathcal{S} \cup \{z_2, z_2'\})\right| \leq Cs^{-(3/2+\upsilon)}.$$

Assumption 1 is a standard regularity condition commonly adopted in the data valuation literature [14, 33], and Assumption 2 introduces a mild second-order stability requirement. In Section 3.5, we will show examples that Assumption 2 is satisfied by some commonly used utility functions.

**Theorem 2.** *Under Assumptions 1 and 2, for each* $s \in \{1, \ldots, n-1\}$, *we have*

$$\mathcal{T}(s) = n/(n-1) \cdot \alpha_0(1 - \alpha_0)\left\{\Delta\mu\left(s_1^*/s; s, s_0, n\right) + O\left(s^{-(1+\upsilon)}\right)\right\}, \tag{9}$$

*where* $s_1^* := \lfloor ss_0/n \rfloor$, $\alpha_0 := s_0/n$, *and* $\Delta\mu\left(\frac{s_1}{s}; s, s_0, n\right) := \mu\left(\frac{s_1+1}{s}; s, s_0, n\right) - \mu\left(\frac{s_1}{s}; s, s_0, n\right)$.

Theorem 2 consolidates Key observation 3 and, moreover, shows that its approximation error decays rapidly as $s$ grows. For large $s$, we can use (9) to design the estimator for $\mathcal{T}(s)$. Yet, some extra care is needed. In principle, each $\mu(s_1/s; s, s_0, n)$ can be estimated by subsampling $S$ from $\mathscr{A}_{s,s_1}$.

$$\widehat{\mu}_m\left(\frac{s_1}{s}; s, s_0, n\right) := \frac{1}{m}\sum_{j=1}^{m} U(S^{(j)}), \quad \text{where } \{S^{(j)}\}_{j=1}^m \overset{\text{i.i.d.}}{\sim} \mathrm{Uniform}(\mathscr{A}_{s,s_1}). \tag{10}$$

However, estimating $\mu(\frac{s_1+1}{s}; s, s_0, n)$ and $\mu(\frac{s_1}{s}; s, s_0, n)$ *separately* can be *statistically inefficient*, as the noise from two independent Monte Carlo estimates could mask the signal in their difference—especially when the true gap is small for large $s$. Following the variance-reduction technique used in stochastic simulation [21], we propose to estimate $\Delta\mu$ directly using *paired* Monte Carlo terms:

$$\widehat{\Delta\mu}_m\left(\frac{s_1}{s}; s, s_0, n\right) := \frac{1}{m}\sum_{j=1}^m \left\{ U\big(S^{(j)}\cup\{i_1^{(j)}\}\big) - U\big(S^{(j)}\cup\{i_2^{(j)}\}\big)\right\}, \qquad (11)$$

where the tuple $\{(S^{(j)}, i_1^{(j)}, i_2^{(j)})\}$ is i.i.d. sampled from $\big\{(S, i_1, i_2) : |S| = s, |S\cap S_0| = s_1, i_1 \in S_0\setminus S, i_2 \in S_0^c\setminus S\big\}$.

When $s$ is small and the approximation becomes less accurate, we instead estimate $\mathcal{T}(s)$ via direct Monte Carlo using (8) and (10).

We formally present our method as Algorithm 1. Later, our Theorem 3 will provide quantitative guidance on choosing this threshold for deciding whether $s$ is small or large.

### 3.3 Computational complexity

Following the convention, we measure computational complexity by the number of utility function evaluations required for a given approximation accuracy.

**Definition 3** (($\epsilon, \delta$)-approximation). *For a target vector $\theta \in \mathbb{R}^d$, an estimator $\widehat{\theta}$ is called an ($\epsilon, \delta$)-**approximation**, if $\mathbb{P}\big(\|\widehat{\theta} - \theta\|_2 \geq \epsilon\big) \leq \delta$.*

Our theoretical analysis for approximating $\mathcal{T}(s)$ relies on the stability of the utility function $U$. Specifically, we employ the concept of *deletion stability* to quantify the maximum change in the utility function when a single data point is removed [1, 11].

---

**Algorithm 1** Approximate FGSV($S_0$)

---

**Require:** Dataset $\mathcal{D}$, group $S_0$, threshold $\bar{s}$, subsample sizes $m_1, m_2$.
1: Initialize $n = |\mathcal{D}|$, $s_0 = |S_0|$ and $\alpha_0 = s_0/n$.
2: **for** $s = 1$ to $n - 1$ **do**
3:     **if** $s < \bar{s}$ **then**
4:        Estimate $\widehat{\mu}_{m_1}(\frac{s_1}{s}; s, s_0, n)$ for each $s_1 \in [\max\{0, s+s_0-n\}, \min\{s, s_0\}]$ by Eq. (10).
5:        Compute $\widehat{\mathcal{T}}(s)$ by (8), replacing $\mu$ by $\widehat{\mu}_{m_1}$.
6:     **else**
7:        $s_1^* \leftarrow \lfloor s\alpha_0\rfloor$.
8:        Estimate $\widehat{\Delta\mu}_{m_2}(\frac{s_1^*}{s}; s, s_0, n)$ by Eq. (11).
9:        $\widehat{\mathcal{T}}(s) \leftarrow \frac{n}{n-1}\alpha_0(1-\alpha_0)\cdot\widehat{\Delta\mu}_{m_2}(\frac{s_1^*}{s}; s, s_0, n)$.
10:    **end if**
11: **end for**
12: **return** $\frac{s_0}{n}[U([n]) - U(\varnothing)] + \sum_{s=1}^{n-1}\widehat{\mathcal{T}}(s)$.

---

**Definition 4** (Deletion Stability). *A utility function $U$ is $\beta(s)$-deletion stable for a non-increasing function $\beta: \mathbb{N} \to \mathbb{R}^+$, if*

$$|U(\mathcal{S}\cup\{z\}) - U(\mathcal{S})| \leq \beta(s),$$

*for all $s \in \mathbb{N}$, $\mathcal{S} \in \mathcal{Z}^{s-1}$ and $z \in \mathcal{Z}$.*

The regime $\beta(s) = O(1/s)$ for deletion stability is commonly assumed in the literature on individual Data Shapley approximation [36, 37] and in the analysis of algorithm stability [1, 11].

**Theorem 3.** *Suppose the utility function $U$ is $O(1/s)$-deletion stable, then Algorithm 1 guarantees that for any truncation threshold $\bar{s}$ and sample sizes $m_1, m_2$, with probability at least $1 - \delta$,*

$$\big|\widehat{\mathrm{FGSV}}(S_0) - \mathrm{FGSV}(S_0)\big| \lesssim \bar{s}\sqrt{\frac{\log(n/\delta)}{m_1}} + \alpha_0(1-\alpha_0)\sqrt{\frac{\log(n/\delta)}{m_2}}\log n + \alpha_0(1-\alpha_0)\bar{s}^{-\upsilon}.$$

*Specifically, choosing*

$$\bar{s} \asymp \epsilon^{-1/\upsilon}, \quad m_1 \asymp \epsilon^{-\frac{4+2\upsilon}{\upsilon}}\log(n/\delta), \quad m_2 \asymp \max\Big\{1, \epsilon^{-2}(\alpha_0(1-\alpha_0))^2(\log(n/\delta))^3\Big\},$$

*yields an ($\epsilon, \delta$)-approximation of $\mathrm{FGSV}(S_0)$ with $O\big(n\cdot\max\big\{1, (\alpha_0(1-\alpha_0))^2(\log n)^3\big\}\big)$ utility evaluations.*

Theorem 3 implies that our algorithm requires only $O(n\,\mathrm{Poly}(\log n))$ utility evaluations to achieve an ($\epsilon, \delta$)-approximation for the FGSV of a group $S_0$ whose size scales as a constant fraction of $n$.

To compare our method's computational complexity with existing works, we notice numerous recent efficient algorithms for approximating individual Shapley values [14, 34, 24, 26, 3, 27, 23, 38]. They target an $(\epsilon, \delta)$-approximation on the full individual Shapley-value vector $\|\widehat{\theta} - \theta\|_2$, where $\theta := (\mathrm{SV}(1), \ldots, \mathrm{SV}(n))$ and $\widehat{\theta}$ is the approximation for $\theta$. The SOTA method achieves this guarantee in $O(n\epsilon^{-2}\log(n/\delta))$ utility evaluations. Approximating $\mathrm{FGSV}(S_0) = \sum_{i \in S_0} \theta_i$ by simply summing up individual approximations can lead to an additive error bounded by $\sqrt{s_0}\|\widehat{\theta} - \theta\|_2$. Thus, achieving the same $(\epsilon, \delta)$-approximation on $\mathrm{FGSV}(S_0)$ requires tightening the error tolerance to $\epsilon/\sqrt{s_0}$, increasing the sample complexity to $O(\alpha_0 n^2 \mathrm{Poly}(\log n))$. By directly targeting the group objective, our method avoids this quadratic blow-up and offers a significant speed-up over the SOTA. It continues to outperform SOTA individual-based methods even when evaluating multiple groups in parallel, provided that the number of groups is $o(n)$.

## 3.4 Utility function values for small input sets

In some machine learning scenarios, $U(S)$ might not be well-defined for small $|S|$. For instance, methods such as LLM's would only produce meaningful result from sufficiently large data sets. Meanwhile, Shapley value emphasizes the contributions from small games, therefore, we cannot simply ignore the small $|S|$ terms in the computation of FGSV (7). One way is to use variants of Shapley value, such as beta-Shapley and Banzhaf, that down-weight small $|S|$ terms, but these alternatives do not satisfy all axioms for faithful group data valuation per Theorem 1. The other way is to fill in $U(S)$ for small $S$ with random or zero values. However, these ad-hoc solutions lack principle and may risk significantly distorting the valuation.

To motivate our approach, we make two simple observations. First, when $S = \emptyset$, the trained method should behave as if it were trained on pure noise, corresponding to a *baseline* utility value. Second, big-data reliant methods might not value small $S$ very differently than baseline, regardless of their content. In this context, both quality and quantity of data matter for valuation.

Our remedy is very simple. We set a threshold for input size, denoted by $B$, such that $U(S)$ is always well-defined and meaningful for $|S| \geq B$. For example, in a linear regression with $p$ predictors, observing that $U(S)$ is undefined for $|S| < p$, we may set $B = cp$ for some constant $c \geq 1$. If $|S| < B$, we inject $B - |S|$ *non-informative* data points, to elevate the input size to $B$; otherwise no change is made to $U(S)$. As a concrete example for the non-informative distribution, consider a supervised learning scenario with data $D = \{(x_i, y_i)\}_{i=1}^n$. Here, we can randomly shuffle $y_i$'s and use the resulting empirical data distribution as the non-informative distribution $\mathcal{P}_{\mathrm{null}}$. Clearly, we can expect that as the size of informative data (i.e., $|S|$) increases, less amount of non-informative data will be injected, and $U(S)$ gradually becomes non-baseline. For further theoretical and algorithmic details, see Appendix C.

While our approach was inspired by the feature deletion technique in *machine unlearning*, to our best knowledge, we are the first to adapt the idea for data Shapley, in a distinct fashion.

## 3.5 Examples of second-order stable algorithms

Among the conditions our theory needs, Assumption 2 is arguably the least intuitive. To demonstrate that Assumption 2 is in fact quite mild, here, we verify that it holds for two important applications.

### 3.5.1 Stochastic Gradient Descent (SGD)

SGD on a training set $\mathcal{S} = \{z_i\}_{i=1}^s$ minimizes the empirical loss $\mathcal{L}(w) = \frac{1}{s}\sum_{i=1}^s \ell(w; z_i)$. Starting from an initial $w_0$, SGD runs for $T$ steps, each step $t \in [T]$ updating $w_t = w_{t-1} - \alpha_t \cdot \frac{1}{m}\sum_{i \in I_t} \nabla\ell(w_{t-1}; z_i)$ with learning rate $\alpha_t$, where the mini-batch $I_t$ of size $m$ is drawn independently and uniformly from $[n]$. Denote the final output by $w_T = w(z_{I_1}, \ldots, z_{I_T})$. The performance of $w_T$ can be represented as some scalar function $u(\cdot)$ (e.g., classification accuracy on a test data). Define the utility as the expected performance:

$$U(\mathcal{S}) := \mathbb{E}_{I_1, \ldots, I_T}[u(w_T)]. \tag{12}$$

**Assumption 3.** *Suppose $\ell$ and $u$ satisfy the following regularity conditions. All conditions hold for some constants $C, L, \beta, \rho > 0$ and all $w, w', z$.*

1. *(Smoothness of $u$) $\|\nabla u(w)\| \leq C$ and $\|\nabla u(w') - \nabla u(w)\| \leq C\|w' - w\|$.*

2. *(L-Lipschitz of $\ell$) $|\ell(w'; z) - \ell(w; z)| \leq L\|w' - w\|$.*

3. *($\beta$-smoothness of $\ell$)* $\|\nabla_w \ell(w'; z) - \nabla_w \ell(w; z)\| \leq \beta \|w' - w\|$.

4. *(Smoothness of $\ell$'s Hessian)* $\|\nabla_w^2 \ell(w'; z) - \nabla_w^2 \ell(w; z)\| \leq \rho \|w' - w\|$.

Assumption 3.1 imposes mild smoothness on the utility function $u$, while the remaining assumptions concern the loss function $\ell$. Compared with the classical analysis of SGD stability by Hardt et al. [11], the only additional requirement is Assumption 3.4, which ensures smoothness of the Hessian. This condition is mild and is satisfied by standard smooth loss functions, such as the squared loss, when the prediction function is twice continuously differentiable in terms of parameters $w$.

**Proposition 2.** *Set* $\alpha_t \asymp s^{-\tau_1}/t$, $m \asymp s^{\tau_2}$, *and* $T \asymp s^{\tau_3}$, *where constants* $\tau_1 > 0$ *and* $\tau_2, \tau_3 \geq 0$. *Suppose* $\upsilon := 2\tau_1 + \tau_2 - 1/2 \geq 0$. *Under Assumption 3, the utility function* (12) *satisfies:*

$$|U(\mathcal{S} \cup \{z_1, z_1'\}) - U(\mathcal{S} \cup \{z_1, z_2\}) - U(\mathcal{S} \cup \{z_1', z_2'\}) + U(\mathcal{S} \cup \{z_2, z_2'\})| \lesssim s^{-(3/2+\upsilon)}.$$

### 3.5.2 Influence Function (IF)

The influence function (IF) method [9, 19] considers the following regularized empirical loss:

$$\widehat{\theta}_\mathcal{S} := \arg\min_\theta \mathcal{L}(\theta; \mathcal{S}) := \arg\min_\theta \left\{ s^{-1} \sum_{z \in \mathcal{S}} \ell(\theta; z) + (\lambda/2) \cdot \|\theta\|_2^2 \right\}. \tag{13}$$

Like in Section 3.5.1, suppose the utility can be written as $U(S) = u(\widehat{\theta}_S)$. Under mild conditions (see Proposition 3), standard IF theory [9, 19] implies that

$$U(\mathcal{S} \cup \{z_1, z_1'\}) - U(\mathcal{S}) \approx s^{-1} \nabla_\theta u(\widehat{\theta}_S)^\top H_{\widehat{\theta}_\mathcal{S}}^{-1} \left\{ \nabla \ell(\widehat{\theta}_\mathcal{S}; z_1) + \nabla \ell(\widehat{\theta}_\mathcal{S}; z_1') \right\} \tag{14}$$

for any $z_1, z_1'$, where $H_{\widehat{\theta}_\mathcal{S}}$ is the Hessian of $\mathcal{L}$ evaluated at $\widehat{\theta}_\mathcal{S}$. This yields the following proposition.

**Proposition 3.** *Suppose* $u$ *is continuously differentiable with $L$-Lipschitz and bounded gradient. Assume that the loss function* $\ell(\theta; z)$ *is convex in $\theta$, three times continuously differentiable with all derivatives up to order three uniformly bounded. Then for any* $z_1, z_1', z_2, z_2' \in \mathcal{Z}$,

$$\left| U(\mathcal{S} \cup \{z_1, z_1'\}) - U(\mathcal{S} \cup \{z_1, z_2\}) - U(\mathcal{S} \cup \{z_1', z_2'\}) + U(\mathcal{S} \cup \{z_2, z_2'\}) \right| \lesssim s^{-2}. \tag{15}$$

Proposition 3 is not surprising in view of (14), as the main terms cancel out on the LHS of (15).

## 4 Experiments

We empirically compare our method (FGSV) to the approach of first computing individual Shapley values using SOTA methods and then summing them to form a group valuation. The results demonstrate the superiority of our method in both speed and approximation accuracy and its faithfulness under shell company attack in applications such as copyright attribution in generative AI and explainable AI. We report the key findings in the main paper and relegate full details to Appendix B.

### 4.1 Approximation accuracy and computational efficiency

In this experiment, we compare our method to the following benchmarks: (1) **Permutation**-based estimator [2, 7] that averages marginal contributions over random data permutations; (2) **Group Testing**-based estimator [14, 34] that uses randomized group inclusion tests to estimate pairwise Shapley value differences; (3) **Complementary Contribution** estimator [38] that uses stratified sampling of complementary coalitions; (4) **One-for-All** estimator [24] that uses weighted subsets and reuses utility evaluations; (5) **KernelSHAP** [26], a regression-based method with locally weighted samples; (6) **Unbiased KernelSHAP** [3], KernelSHAP with a bias-correction; and (7) **LeverageSHAP** [27] that speeds up KernelSHAP by weighted sampling based on leverage scores.

**Experimental setup.** We consider the Sum-of-Unanimity (SOU) cooperative game [24], where the individual Shapley values have a closed-form expression. Specifically, the utility function is defined as $U(S) = \sum_{j=1}^d \alpha_j \mathbf{1}_{\mathcal{A}_j \subseteq S}$, and the corresponding Shapley value for player $i \in [n]$ is given by $\mathrm{SV}(i) = \sum_{j=1}^d \frac{\alpha_j}{|\mathcal{A}_j|} \mathbf{1}_{i \in \mathcal{A}_j}$. Here, each subset $\mathcal{A}_j$ is generated by sampling a random size uniformly from 1 to $n$ and then drawing that many players without replacement. We set $d = n^2$ and the coefficient $\alpha_j$ as the average of the weights of all players $i \in \mathcal{A}_j$, where each player $i$ has a weight $\frac{(i \bmod 4)}{4}$. We

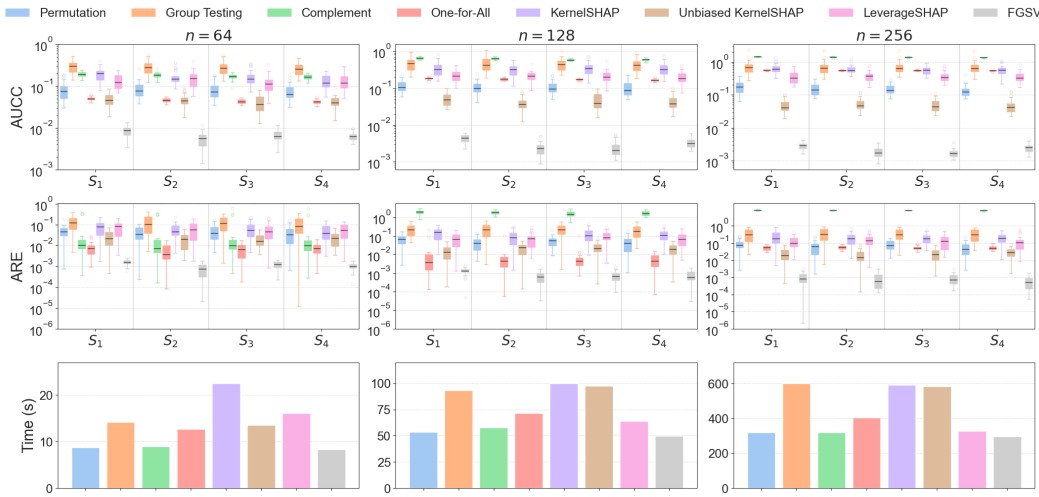

Figure 2: Performance comparison in the SOU game. Top: Our method (FGSV) achieves the lowest AUCC and ARE across all problem sizes. Bottom: Our method costs the lowest runtime per iteration.

partition all players into 4 groups based on their indices: $S_k = \{i \in [n] : i \equiv k - 1 \pmod 4\}$ for $k \in [4]$, and our goal is to estimate the group value $\text{FGSV}(S_k) = \sum_{i \in S_k} \text{SV}(i)$. Given a fixed budget of 20,000 utility function evaluations, we record the absolute relative error of the FGSV estimate every 200 iterations for each method. We summarize convergence behavior using the *Area Under the Convergence Curve* (AUCC), defined as: $\text{AUCC}(S_k) = \frac{1}{100} \sum_{t=1}^{100} |\frac{\text{FGSV}(S_k) - \widehat{\text{FGSV}}^{(200 \cdot t)}(S_k)}{\text{FGSV}(S_k)}|$, which captures both speed and stability of convergence. We also report the *Absolute Relative Error* (ARE) of the final estimate: $\text{ARE}(S_k) = \left| \frac{\text{FGSV}(S_k) - \widehat{\text{FGSV}}^{(20000)}(S_k)}{\text{FGSV}(S_k)} \right|$. Lower AUCC and ARE indicate faster convergence and better accuracy, respectively. We record the average runtime per iteration for each method.

**Results.** Figure 2 summarizes the average performance over 30 replications for $n \in \{64, 128, 256\}$. The top two rows together indicate that our method shows overall superior performance across all problem sizes $n$, exhibiting both the lowest average AUCC and ARE. The bottom row suggests that, while using the same utility evaluation budget, our method is among the fastest. In contrast, baselines such as Group Testing, KernelSHAP, and Unbiased KernelSHAP incur substantial higher computational overhead, due to internal optimization processes. Overall, our method achieves both faster convergence and improved computational efficiency, which demonstrates the benefits of directly estimating the group value rather than aggregating individual Shapley estimates.

## 4.2 Application to faithful copyright attribution in generative AI

Group data valuation is important for fairly compensating copyright holders whose data are used to train generative AI models. Existing approaches, such as the Shapley Royalty Share (SRS) proposed by [35], adopt the GaI approach based on GSV. Given a partition of the training data into $K$ disjoint groups $S_1, \ldots, S_K$, SRS is defined as $\text{SRS}(S_k; x_{\text{gen}}) := \frac{\text{GSV}(S_k; x_{\text{gen}})}{\sum_{j=1}^{K} \text{GSV}(S_j; x_{\text{gen}})}$. However, as discussed in Section 2, GSV is prone to the shell company attack. To address this vulnerability, we propose the Faithful Shapley Royalty Share (FSRS) that replaces GSV with our FGSV to faithfully reward individuals in a group by their contributions, not by tactical grouping strategies:

$$\text{FSRS}(S_k; x_{\text{gen}}) := \frac{\text{FGSV}(S_k; x_{\text{gen}})}{\sum_{j=1}^{K} \text{FGSV}(S_j; x_{\text{gen}})}.$$

**Experimental setup.** Following [35], we fine-tune Stable Diffusion v1.4 [28] using Low-Rank Adaptation (LoRA; [13]) on four brand logos from FlickrLogo-27 [16]. The utility $U(\cdot; x^{(\text{gen})})$ is the average log-likelihood of generating 20 brand-specific images $x^{(\text{gen})}$ using the prompt "A logo by [brand name]" (see example images in Panel (a) of Figure 3). We compare SRS and FSRS under

two grouping scenarios: (1) 30 images from each brand form a single group, and (2) the Google and Sprite datasets are each split into two subgroups (20/10 images), launching a shell company attack. Importantly, the total data per brand remains unchanged across two scenarios.

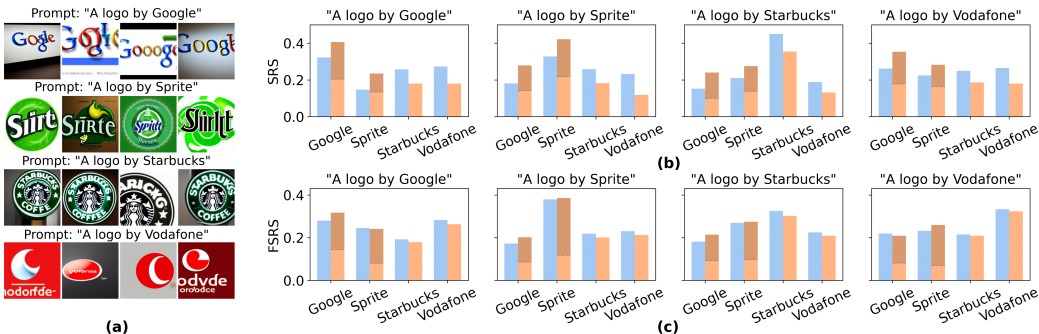

Figure 3: Comparison of SRS and FSRS for copyright attribution. **(a)** Example images generated using brand prompts. **(b)** Shapley Royalty Share (SRS, [35]) based on GSV. **(c)** Faithful SRS (FSRS, our method) based on FGSV. Blue bars: valuation under Scenario 1 (single group per brand); orange bars: valuation under Scenario 2 (Google/Sprite data each split into size-20/10 subgroups, colored in dark and light orange).

**Results.** Figure 3 illustrates the impact of the shell company attack on copyright attribution under SRS and FSRS. Each subplot corresponds to a brand-specific prompt and shows the royalty shares assigned to each brand. Conceptually, the brand corresponding to the prompt should receive the highest royalty share. Panel (b) shows that applying a shell company attack that favors the Google and Sprite groups indeed inflates their total SRS shares substantially, while reducing others', despite the contents contributed by each group remain unchanged. This produces misleading results—for example, Google and Sprite receive higher overall SRS than Vodafone under the "`A logo by Vodafone`" prompt. In contrast, Panel (c) shows that FSRS yields consistent and stable valuations under the shell company attack. This demonstrates that FSRS mitigates the effects of strategic data partitioning and provides a more faithful reflection of group contribution for copyright attribution.

## 4.3 Application to faithful explainable AI

Group data valuation is also a crucial tool in explainable AI, providing interpretable summaries of data contributions at the group level— particularly in contexts where groups correspond to socially or scientifically significant categories. For example, [7] used GSV to quantify the contributions of data from different demographic groups to patient readmission prediction accuracy [31]. However, GSV's sensitivity to the choice of grouping can cause group values to fluctuate dramatically under different data partitions, leading to inconsistent interpretations. To address this problem, we recommend practitioners to use FGSV and empirically demonstrate that it produces more consistent and reliable interpretations across a variety of grouping configurations.

**Experimental setup.** We conduct our experiment on the Diabetes dataset [5], which contains 442 individuals, each described by 10 demographic and health-related features (e.g., *sex*, *age*, and *BMI*). The task is to predict the progression of diabetes one year after baseline. We construct 7 grouping schemes, partitioned by all non-empty subsets of the variables *sex*, *age*, and *BMI*, excluding the trivial no-partition case. For the continuous variables *age* and *BMI*, we discretize each into three quantile-based categories. Overall, we will have between 2 to 18 total groups. For each grouping scheme, we compute GSV exactly and estimate FGSV via 30 Monte Carlo replications. Our predictive model is ridge regression, and we measure utility as the *negative* mean squared error on a held-out test set, with the null utility set to the variance of the test responses. Thus, higher group valuations indicate greater contributions to the model's predictive accuracy. To compare category-level contributions across different grouping schemes, we aggregate group values as follows: for a given variable (e.g., *sex*), we sum the (F)GSVs of all groups that include that category. This allows for consistent cross-scheme comparisons, where one would expect a category to receive stable valuations after aggregation across different grouping strategies if the valuation method is robust.

**Results.** Figure 4 visualizes the aggregated category-level values from GSV (top row) and FGSV (bottom row) across various grouping schemes. FGSV produces significantly more stable and

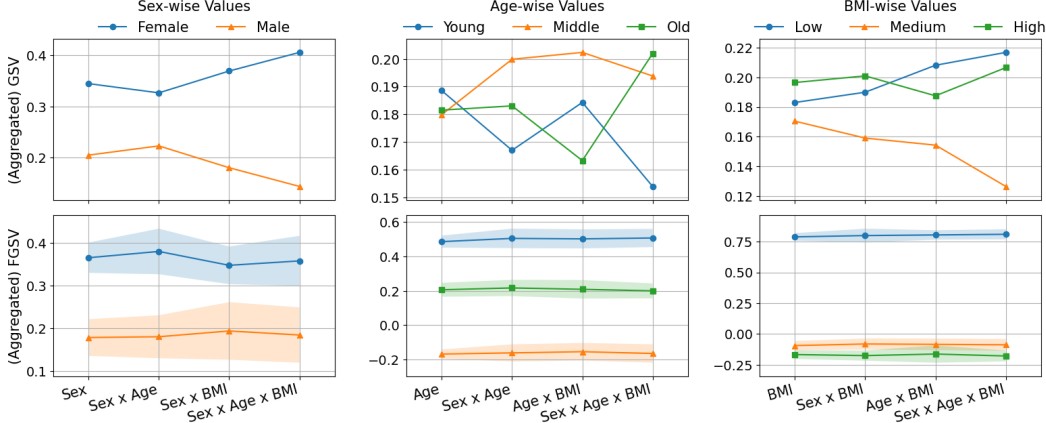

Figure 4: Comparison of GSV (top row) and FGSV (bottom row) in a regression task for explainable AI. Each column aggregates category-level values for a specific variable: *sex* (left), *age* (middle), and *BMI* (right). Shaded areas represent $\pm 1$ standard deviation across 30 replications.

consistent group rankings compared to GSV. For instance, in the *age*-wise plots (second column), GSV gives inconsistent rankings: each of the three age groups—"Young," "Middle," and "Old"— appears as the top-ranked group under at least one grouping scheme. This instability undermines the reliability of the interpretation. In contrast, FGSV consistently assigns the highest value to the "Young" group across all grouping schemes, demonstrating its robustness. Overall, FGSV provides more reliable explanations that are less sensitive to arbitrary grouping schemes.

## 5 Conclusion and discussion

In this paper, we proposed FGSV for faithful group data valuation. We showed that FGSV is the unique group valuation method that satisfies a desirable set of principles, including faithfulness, which ensures that group value remains unchanged to arbitrary re-grouping among other players, thereby defending against the shell company attack. Our algorithm also achieves lower sample complexity over SOTA methods that sum up individual Shapley value estimates, as demonstrated through both theoretical analysis and numerical experiments. We further illustrated the robustness of FGSV in applications to copyright attribution and explainable AI, where it faithfully reflects and fairly rewards individual contributions from group members.

Beyond the shell company attack, there exists another unfair competition strategy, namely, the **copier attack**, in which a group may steal valuation from other groups by duplicating their high-value data points. An ad-hoc remedy is to pre-process the dataset to detect and remove such duplicated entries before applying our FGSV method. In existing literature, [6] suggests designing the utility function using Pearl's "do operation". This defense can be incorporated into our framework, letting our method also defend against the copier attack; but such do-utility function is not always available. [10] shows that with a submodular utility function, Banzhaf [33] and leave-one-out can successfully discourage a genuine contributor from duplicating itself to gain higher total valuation; however, their method is prone to "pure infringers" who only copy from valuable data points without contributing original contents. Overall, it remains an open challenge to defend against the copier attack for general utility functions, while maintaining the defense against the shell company attack to ensure a safe group data valuation.

## Code

The code and instructions to reproduce the experiments are provided in the supplementary material and available at `https://github.com/KiljaeL/Faithful_GSV`.

## Acknowledgements

Lee and Zhang were supported by NSF DMS-2311109. Liu and Tang were supported by NSF DMS-2412853.

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
