# OpenReview forum: "Faithful Group Shapley Value"
_NeurIPS.cc/2025/Conference — NeurIPS 2025 poster_

### Official Review · Reviewer_R2x5 · 2025-06-17

**Clarity:** 2
**Significance:** 3
**Originality:** 3
**Rating:** 4
**Confidence:** 4

**Summary:**

The paper considers the problem of group data valuation and proposes axioms that a _faithful_ group data valuation method should satisfy. In particular, for faithfulness, the total valuation of the same group of data should remain unchanged, regardless of how others are partitioned. The paper suggests the sum of individual Shapley values (i.e., the Faithful Group Shapley Value (FGSV)) uniquely satisfies the required axioms.
Next, based on the observation that a small subset of terms dominate in FGSV’s formula, the paper proposes a more efficient approximation for individual FGSV.
Lastly, the paper empirically evaluate the approximation accuracy and computation time of FGSV and other baseline methods and compare FGSV against GSV on copyright attribution in generative AI application and explainable AI applications.

**Questions:**

1. What is the justification for the weaker null player and symmetry axioms in Sec 3 vs their GaI counterparts? Why is this weakening needed or acceptable?
2. In line 193-6, it is stated that SOTA individual data values require quadratic number of samples to achieve the same error tolerance. However, the quadratic number of samples (or more precisely linear multiplied by the maximum group size) can be reused across all groups $S_0, \dots, S_k$. Algorithm 1 requires linear number of samples to achieve the error tolerance but it is only for a specific group $S_0$. With more groups (maximum $n$), the total number of samples becomes quadratic as well. Is this understanding correct? If so, is the new approximation less useful and efficient in some cases with more groups?
3. What is the issue with small input sets (Sec 3.4)? For smaller datasets, the utility might be small but that does not mean it is meaningless. Do you mean that the variance due to the algorithm randomness is high?
4. In Sec 4.1, it is good that the total budget of utility evaluations is fixed at 20k. However, comparing the performance and time taken based on the iteration of each method seems unfair as _different methods use different number of utility evaluations per iteration_ (e.g., permutation vs single utility). Theorem 3 suggests a comparison based on the same number of utility evaluations instead.
    * Can you compare based on the number of utility evaluations (plot of AUCC/MSE against number of utility evaluations)?
    * Can you report the total runtime instead? Is the runtime for a single FGSV or all $n$ FGSVs?

**Ethical Concerns:**

["NO or VERY MINOR ethics concerns only"]

**Final Justification:**

The authors' response has addressed my question and concerns. I have raised the score slightly.

**Limitations:**

The authors only discussed another unaddressed attack strategy in the conclusion.

**Quality:**

2

**Strengths And Weaknesses:**

### Strengths
1. It is significant and important to design group data valuation that satisfy the faithfulness axiom and is resistant to the splitting and shell company attacks. While summing the individual Shapley values seem like a natural solution, the paper also proposes a novel technique to compute the FGSV more directly and efficiently.

### Weaknesses
1. The clarity of the paper can be improved.
    * A few claims need some clarifications (see questions below)
    * Some notations are unexplained, e.g., $\asymp$ in Theorem 3, $\mathcal{P}\^s$ on line 77, $\mathcal{Z}\^s$ in Assumption 1 and 2. Is  $\mathcal{P}^s$ the uniform distribution over all subsets of size $s$?
    * It would be helpful to provide the intuitive version and justification of the prudence condition (Eq 3) and axioms in Sec 3 so that the reader would not have to infer them. It is also helpful to provide some intuition about the proof of the proposition and the theorem in the main paper.
2. There are some concerns about the experiment settings and whether it demonstrates FGSV better efficiency and approximation quality over other individual SV estimation methods (see questions below)

### Minor Comments
* Using $s_1$ instead of $s_1/s$ in the definition of $\mu$ (Eq. 6) may be cleaner. $\mu$ depends on the set $S_0$ not just the size $s_0$.
* For the copier/replication attack, see Han, D., Wooldridge, M., Rogers, A., Ohrimenko, O., & Tschiatschek, S. (2022). Replication robust payoff allocation in submodular cooperative games. IEEE Transactions on Artificial Intelligence, 4(5), 1114-1128.

---

> ### Author Rebuttal · Authors · 2025-07-31
>
> Thank you for your thoughtful and constructive comments.
>
> > **1. Notation clarification and improvement suggestions (minor comment 1)**
>
> We will improve clarity in the revised paper.
> - "$\asymp$" is imported from calculus to denote asymptotic equivalence: $a_n\asymp b_n$ means both $a_n=O_p(b_n)$ and $b_n=O_p(a_n)$. Here, $a_n=O_p(b_n)$ means that the sequence $a_n/b_n$ is bounded in probability.
> - "$\mathcal{P}^s$" denotes the product distribution of $s$ independent samples from the distribution $\mathcal{P}$, i.e., $S\sim\mathcal{P}^s$ if $S=(z_1,\dots,z_s)$ with $z_i\stackrel{\mathrm{iid}}{\sim}\mathcal{P}$.
> - Similarly, "$\mathcal{Z}^s$" denotes the Cartesian product of $\mathcal{Z}$ taken $s$ times, i.e., $S\in\mathcal{Z}^s$ if $S=(z_1,\dots,z_s)$ with each $z_i\in\mathcal{Z}$.
> - Following your suggestion, we will revise the paper using $s_1$ in Eq (6).
>
> > **2. How to understand the prudence condition?**
>
> Prudence is a classical concept in economics: a consumer is called “prudent” if they save more when facing greater future income risk – mathematically, this is characterized as the utility’s third derivative being nonnegative [4].  In machine learning applications, several popular utility function curvatures satisfy the prudence condition, such as
> - Power‑law utilities ($\bar U(s)\propto s^a$ with $0<a<1$)
> - Logarithmic utilities ($\bar U(s)\propto \log s$)
> For instance, empirical studies report that scaling laws in deep learning [2] and large‑language‑model performance [3] follow power‑law trends in sample size \(s\), thereby also satisfying the prudence condition.
>
> > **3. How to interpret the axioms in Definition 2? Why use weaker null player and symmetry axioms, compared against their GaI counterparts, why is this needed or acceptable?**
>
> Our five axioms closely mirror the classical Shapley axioms, with only minimal adaptations to enforce faithfulness against shell attacks. We will add references, include a concise recap in the appendix, and provide clear explanations in the main text so readers can readily understand each axiom’s intent in the revised version.
>
> The weaker null player and symmetry axioms are both mathematically necessary and conceptually sensible.
>
> **Mathematical necessity.**  Roughly speaking, we can think of axioms as “assumptions” or “requirements”.  Stronger axioms make it harder to find compliant valuation methods.  To fence off the shell company attack, the valuation method must satisfy our newly identified faithfulness axiom.  Then, if one still enforces the GaI version of axioms 1–4, no method can comply with all five axioms.
>
> To illustrate, consider the following toy example:
>
> **Players** $\\{1,2,3\\}$ are fully interchangeable but **do not collaborate**: the utilities are
> $$
> U(S)= \begin{cases} 0, & S=\varnothing,  \\\ 1, & S\neq\varnothing. \end{cases}
> $$
> In other words, adding a second or third player to a nonempty set brings **no extra utility** beyond what a single player already provides.
>
> **Grouping** $A=\\{1,2\\}$ vs. $B=\\{3\\}$.  Under GaI‑Symmetry, groups $A$ and $B$ are each treated as atomic players.  Symmetry requires that if
> $$
> U(S\cup A)=U(S\cup B)\text{ for all }S\subseteq \\{1,2,3\\}\setminus(A\cup B),
> $$
> then they must receive equal value, i.e. $\phi(A)=\phi(B)$. Here the only external coalition is $S=\varnothing$, and $U(A)=U(B)=1$, so GaI‑Symmetry forces $\phi(A)=\phi(B)$.
>
> **This fails to comply with axiom 5.** Since each atomic player contributes the same marginal utility (1 when first added, 0 thereafter), a faithful valuation should assign $A$ twice the value of $B$, reflecting its two members. The GaI axioms obscure this intra‑group structure, leading to the mismatch.
>
> **Conceptual sensibility.**  The GaI paradigm treats each group as an indivisible “super‑player,” ignoring interactions and unique contributions from its members.  This is partially why GaI is vulnerable to the shell company attack. Therefore, to develop a more robust valuation method, null player and symmetry must be revised to reflect interactions between individuals, ensuring that grouping does not cut off these interactions as it does in GaI.
>
>
> > **4. Provide proof sketches**
>
> Thank you for the suggestion. In the revised appendix, we will include a high‑level outline of the main ideas at the beginning of each major proof to help readers follow the reasoning without wading through every technical detail.
>
> > **5. Related work on copier attack [1]**
>
> Thank you for suggesting this paper. We will certainly cite this work in our revised paper. This paper establishes that when using a submodular utility function, Banzhaf and leave-one-out can automatically demotivate self-duplication from genuine contributors, while Shapley does not.  It successfully addresses a key subtype of copier attack. Other subtypes of copier attack may require different solutions. For example, the challenge of "pure infringers", where new entrants who contribute nothing but copy valuable data from others, remains an open problem. It would be an interesting future direction to investigate how to simultaneously protect against both the broader class of copier attacks and the shell company attack that is the focus of our work.
>
> > **6. SOTA methods for individual SV produce approximations for all $\mathrm{SV}(i)$’s, while this paper’s algorithm for FGSV only computes for one $\mathrm{SV}(S_0)$.  What if there are multiple $S_0$’s to evaluate – is it still faster?**
>
> Yes, our method remains faster than individual‑Shapley approximations as long as the number of groups $K$ grows as $o(n)$. Ignoring logarithmic factors, existing individual-SV-based methods require $O(n^2)$ utility evaluation to compute all SV$(S_k)$ terms for $k \in \\{1, \cdots, K\\}$. In contrast, our method requires only $O(n)$ utility evaluations to compute a single $\mathrm{SV}(S_0)$, so applying it to $K$ groups costs $O(Kn)$, strictly better than $O(n^2)$ as long as $K=o(n)$.
>
> When there are O(n) groups with balanced group sizes, the scenario is usually not noticeably different than an individual data valuation problem.  In this case, we recommend using individual‑Shapley approximations.
>
> > **7. Clarify what’s the issue for small input sets (Section 3.4).**
>
> First, we clarify that in applications where $U(S)$ is well-defined for small input sets, we do not engage the padding procedure in Section 3.4.  For instance, in Section 4.2, $U(S)$ describes the performance of a foundation model fine-tuned on a small data set $S$.  Since the model is already pretrained, even an empty $S$ gives meaningful $U(S)$. We did not enact Section 3.4 in this example.
>
> Next, we provide some concrete examples where this padding procedure may be useful.
> - Let the utility be the residual norm of standard linear regression with $p$ predictors, then U(S) is not well-defined for $|S|<p$.
> - Another example is that modern learning methods, such as DNN, typically require big data and do not produce informative learning outcomes if trained on small data – in this scenario, the utility, as you mentioned, is unstable and untrustworthy (think of a classification method trained based on 10 observations).
>
> In these examples, by injecting non-informative data to complement small $S$, we stabilize $U(S)$ in a meaningful way and ensure that $U(S)$ for small $S$ is close to the baseline $U(\emptyset)$, which itself also requires a meaningful “padding” in some examples.
>
> > **8. Clarify the computational cost comparison in experiment 1 (Section 4.1)**
> You commented: “Can you compare based on the number of utility…” and “Can you report the total runtime instead?”
>
> Thank you for asking. We would like to clarify a possible misunderstanding: in line 268, we wrote that “recorded the FGSV estimate every 200 iterations,” but the term “iterations” was intended to mean “utility calls”. Therefore, each method was executed with the same total number of utility calls, 20,000, ensuring a fair comparison across all estimators at each evaluation point. The AUCC plots are already generated with the same number of utility evaluations for a direct comparison consistent with Theorem 3.  Regarding runtime, the reported value corresponds to the total runtime required to compute the FGSV’s for all groups. We will improve clarity and avoid potential confusion in the revised version.
>
> ---
> _**References**_
>
> [1] Han, D., Wooldridge, M., Rogers, A., Ohrimenko, O. and Tschiatschek, S., 2022. Replication robust payoff allocation in submodular cooperative games. IEEE Transactions on Artificial Intelligence, 4(5), pp.1114-1128
>
> [2] Hestness, Joel, Sharan Narang, Newsha Ardalani, Gregory Diamos, Heewoo Jun, Hassan Kianinejad, Md Mostofa Ali Patwary, Yang Yang, and Yanqi Zhou. 2017. “Deep Learning Scaling Is Predictable, Empirically.” arXiv [Cs.LG].
>
> [3] Kaplan, Jared, Sam McCandlish, Tom Henighan, Tom B. Brown, Benjamin Chess, Rewon Child, Scott Gray, Alec Radford, Jeffrey Wu, and Dario Amodei. 2020. “Scaling Laws for Neural Language Models.” arXiv [Cs.LG].
>
> [4] Sandmo, A. 1970. “The Effect of Uncertainty on Saving Decisions.” The Review of Economic Studies 37 (3): 353.

---

> > ### Comment · Reviewer_R2x5 · 2025-08-04
> >
> > Thank you for your detailed response that addressed my concerns! I have raised the score.

---

> > > ### Author Response · Authors · 2025-08-04
> > >
> > > Thank you very much!  We really appreciate it.

---

### Official Review · Reviewer_ct4R · 2025-06-27

**Clarity:** 2
**Significance:** 3
**Originality:** 3
**Rating:** 4
**Confidence:** 3

**Summary:**

Authors propose the Faithful Group Shapley Value (FGSV) to ensure data valuation is robust to the shell company attack where different data groups (companies) strategically split their data into multiple groups to increase their value.
They define FGSV for a group as the sum of individual Data Shapley values of its members and prove that it uniquely satisfies the set of axioms desirable for faithful group data valuation.
They empirically test the proposed approach on synthetic data,  FlickrLogo-2 dataset, and diabetes dataset.

**Questions:**

- Section 3.2 (Algorithm 1) in the main paper and in Appendix C.
   - None of the references in Algorithm 1 take you to the actual equation.
   - How do you determine m1 and m2? From B.4, it's 1000>n, and in application B.2, it's 20,000, and in B.1, it's 2000. Is it the same across runs for each application?
   - What is non-informative distribution Pnull, and how does it relate to D?
   - In general, I think that adding a paragraph describing the algorithm(s) and or comments in the algorithm(s) would improve readability.


- Section 3.4 (Utility function values for small input sets).
   - Theorem 3 appears to guide whether the size of s is appropriate; however, it is still unclear to me how B is determined. It's also unclear to me the extent to which the "non-informative data points" affect the data valuation of a group's data. How do authors define non-informative data points? Where do they come from, and are they drawn from the same distribution?
   - Lastly, more theoretical analysis of the implications of this padding process would improve trust in the efficacy of the faithful data valuation approach.

   - The Banzhaf value is replication-robust, reliable across runs, and can handle varied sizes. Why is not used a base method instead of Shapley?



- Experiments (Section 4).
   - Both the experimental setups and the results are generally difficult to follow. For instance, in Figure 2, it is unclear which dataset was used, what the B or \tilde{s} is, what the size of each data group(S) is, and how the dataset was sourced or generated. Many crucial details are missing, with only a few relegated to Appendix B.1. There is also a mismatch between the motivation, e.g., copyright or explainable AI, and the setup and results.
   - The authors mention the test set for just one experiment in Section B.4 but provide no information on the data splits used in the other experiments. It remains unclear how they sourced the held-out test sets and how similar or different they are in distribution compared to the various groups used in the experiments.


- Related works.
   - There are several replication robust "base" methods. For example https://openreview.net/pdf?id=iTjSqQQ4f8, https://proceedings.mlr.press/v206/wang23e/wang23e.pdf, https://arxiv.org/pdf/2006.14583
   - I think the related works should ideally cite papers in this line of inquiry. If possible, use these kinds of methods in the experimental comparative analysis.

- Motivation and setup.
   - There are some disconnects between the motivation and the setup. For example, drawing held-out tests and the different groups from the same distribution might not sufficiently depict the real-world setting the authors aim to address.
   - The application of the approach to faithful explainable AI (section 4.3) might exacerbate fairness issues and further marginalize already marginalized groups. It's also unclear to me, based on the results in Section 4.3, if FGSV improves things.


- Typos and reference issues.
   - There are several typos. For example,  the on the reference made on line 214.  In appendix C, there is only Algorithm 1, not Algorithm 2
   - Most of the references in the paper don't work as expected. E.g., if you click on some hyperlinked references, they take you to page one. For example, none of the references in algorithm 1 take you to the actual equation.

**Ethical Concerns:**

["NO or VERY MINOR ethics concerns only"]

**Final Justification:**

Although the paper has weaknesses, the authors propose an interesting problem supported by sound theory and empirical results. I am raising the score with the expectation that authors will make the promised changes.

**Limitations:**

The paper lacks a (detailed) limitation section

**Quality:**

2

**Strengths And Weaknesses:**

**Strengths**
- The authors try to address a key issue in data markets by using a commonly used method as a baseline.
- Figure 1 is intuitive and clearly illustrates the problem the authors are trying to address.
- Authors theoretically prove the existence of a fast and accurate approximation algorithm for FGSV, empirically test their approach, and avail their code in supplementary material.


**Weaknesses**

- Section 3.2 (Algorithm 1) in the main paper  and in Appendix C.
   - None of the references in Algorithm 1 take you to the actual equation.
   - How do you determine m1 and m2? From B.4, it's 1000>n, and in application B.2, it's 20,000, and in B.1, it's 2000. Is it the same across runs for each application?
   - What is non-informative distribution Pnull, and how does it relate to D?
   - In general, I think that adding a paragraph describing the algorithm(s) and or comments in the algorithm(s) would improve readability.


- Section 3.4 (Utility function values for small input sets).
   - Theorem 3 appears to guide whether the size of s is appropriate; however, it is still unclear to me how B is determined. It's also unclear to me the extent to which the "non-informative data points" affect the data valuation of a group's data. How do authors define non-informative data points? Where do they come from, and are they drawn from the same distribution?
   - Lastly, more theoretical analysis of the implications of this padding process would improve trust in the efficacy of the faithful data valuation approach.

   - The Banzhaf value is replication-robust, reliable across runs, and can handle varied sizes. Why is not used a base method instead of Shapley?



- Experiments (Section 4).
   - Both the experimental setups and the results are generally difficult to follow. For instance, in Figure 2, it is unclear which dataset was used, what the B or \tilde{s} is, what the size of each data group(S) is, and how the dataset was sourced or generated. Many crucial details are missing, with only a few relegated to Appendix B.1. There is also a mismatch between the motivation, e.g., copyright or explainable AI, and the setup and results.
   - The authors mention the test set for just one experiment in Section B.4 but provide no information on the data splits used in the other experiments. It remains unclear how they sourced the held-out test sets and how similar or different they are in distribution compared to the various groups used in the experiments.


- Related works.
   - There are several replication robust "base" methods. For example https://openreview.net/pdf?id=iTjSqQQ4f8, https://proceedings.mlr.press/v206/wang23e/wang23e.pdf, https://arxiv.org/pdf/2006.14583
   - I think the related works should ideally cite papers in this line of inquiry. If possible, use these kinds of methods in the experimental comparative analysis.

- Motivation and setup.
   - There are some disconnects between the motivation and the setup. For example, drawing held-out tests and the different groups from the same distribution might not sufficiently depict the real-world setting the authors aim to address.
   - The application of the approach to faithful explainable AI (section 4.3) might exacerbate fairness issues and further marginalize already marginalized groups. It's also unclear to me, based on the results in Section 4.3, if FGSV improves things.


- Typos and reference issues.
   - There are several typos. For example,  the on the reference made on line 214.  In appendix C, there is only Algorithm 1, not Algorithm 2
   - Most of the references in the paper don't work as expected. E.g., if you click on some hyperlinked references, they take you to page one. For example, none of the references in algorithm 1 take you to the actual equation.

---

> ### Author Rebuttal · Authors · 2025-07-31
>
> Thank you for your thoughtful and constructive comments.
>
> > **1. Typos, broken hyperlinks, and adding comments and description to algorithms**
>
> We will correct the typos/hyperlinks, and add comment lines into algorithms to improve readability in the revised version. Algorithm 1 in the Appendix should be Algorithm 2 for an optional padding procedure when the utility function is not well-defined for small sets.
>
> > **2. Clarification on the choice of tuning parameters $m_1$ and $m_2$ in Algorithm 1**
>
> While Theorem 3 provides a theoretical guideline for accuracy, in practice, we selected $m_1$ and $m_2$ to balance approximation quality and computational cost.
> Specifically, Theorem 3 implies an error bound of
> $$O\bigg(\frac{\log n}{m_1}+\frac{\log s}{m_2}\bigg),$$
> which decreases with larger $m_1$ and $m_2$. However, since both $m_1$ and $m_2$ are the numbers of Monte Carlo samples, increasing them leads to more utility calls, roughly $s_0m_1+2m_2$ calls, which directly affects total runtime. Importantly, different experiments incur utility costs differently, depending on the model complexity and sample size. Thus, in each experiment, we selected $m_1$ and $m_2$ such that the total utility calls remained within the computational budget, while accounting for the per-call cost in specific settings.
> - B.1: Involves retraining with low-dimensional input and small sample size, allowing large $m_1 = m_2 = 2000$.
> - B.4: Involves retraining with high-dimensional input and large sample size, setting $m_1 = m_2 = 1000$ to keep the runtime manageable.
>
> In B.2, the number 20,000 refers to the total number of utility calls allowed for each method, not to fixed $m_1$ and $m_2$. Here, utility does not involve retraining, so we could afford to set $m_1$ and $m_2$ adaptively for each $n$ to match this budget constraint.
>
> > **3. The padding procedure in Section 3.4**
>
> **i) How to select $B$?**
> The choice of $B$ depends on the specific context. $B$ denotes the smallest $s$ for which $U(S)$ is well-defined. For example, in linear regression with $p$ predictors, $U(S)$ is undefined for $|S|<p$. In this case, we typically set $B=cp$ for some constant $c\geq 1$.
>
> Also, let us clarify that when $U(S)$ is well-defined for small input sets, we do not engage the procedure in Section 3.4. For instance, in Section 4.2, $U(S)$ describes the performance of a foundation model fine-tuned on a small data set $S$. Since the model is already pretrained, even an empty $S$ gives meaningful $U(S)$. We did not enact Section 3.4 in this example.
>
> **ii) How to generate the non-informative distribution?**
>
> In supervised learning, with data $D = \\{(x_i,y_i)\\}^n_{i=1}$ we can randomly shuffle the order of $y_i$'s, then the empirical distribution of the shuffled data can serve as $\mathcal{P}_{null}$.
>
> **iii) Theoretical analysis of this padding procedure**
>
> We will include a full theoretical analysis of the padding procedure for Algorithm 2 in the revised Appendix. Here, we briefly present the main result. First, define a padded utility function
> $$
> \tilde U(S)=\begin{cases}\mathbb{E}\bigl[U(S\cup S_{null})\bigr],&|S|<B,\\\U(S),&|S|\ge B, \end{cases}
> $$
> where $S_{null}\sim \mathcal{P}_{null}^{B-|S|}$.
>
> Under the assumption that $U$ satisfies Assumptions 1 and 2 and is $O(1/s)$‑deletion‑stable for $|S|\ge B$, one can verify that $\tilde U$ retains these properties for all $S$. Applying Algorithm 2 to $\tilde U$ with the parameter choices
>
> $$\bar s\asymp\epsilon^{-1/\upsilon}\bigl(\alpha_0(1-\alpha_0)\bigr)^{1/\upsilon}\log^{1/(2\upsilon)}\bigl(n/\delta\bigr),\quad m_1\asymp\epsilon^{-\frac{2(2+\upsilon)}{\upsilon}}\bigl(\alpha_0(1-\alpha_0)\bigr)^{4/\upsilon}\log^{\frac{2+\upsilon}{\upsilon}}\bigl(n/\delta\bigr),\quad m_2\asymp\epsilon^{-2}\bigl(\alpha_0(1-\alpha_0)\bigr)^2\log^3\bigl(n/\delta\bigr),$$
>
> delivers an $(\epsilon,\delta)$‑approximation of $\widetilde{\mathrm{FGSV}}(S_0)$ using
>
> $$O\bigl((\alpha_0(1-\alpha_0))^2n(\log n)^3\bigr)$$
>
> utility evaluations, matching the bounds in the original Theorem 3.
>
> > **4. Our motivation and problem setup are orthogonal to machine learning fairness.** You commented:
> “drawing held-out tests and the different groups from the same distribution...”
> and
> “The application of the approach to faithful explainable AI (section 4.3) might exacerbate fairness...”
>
> While we agree that ML fairness is important, it concerns a different aspect and is **orthogonal to the main theme of this paper.**
>
> A Shapley-type data valuation method typically has two stages: 1. Design a utility function $U(\cdot)$; 2. Properly combine $U(\cdot)$ to yield an individual or group valuation measure.
>
> Our study focuses on stage 2, where ‘faithful’ refers to robustness against the shell company attack, which provably compromises GSV-based data valuation.
>
> We understand that your comments entail adjusting the prediction to reduce disparity among different subgroups. Designing the prediction method and evaluating its outcome both belong to stage 1 and eventually reflected in $U(\cdot)$. In contrast, we focus on the valuation faithfulness in stage 2 for any given $U(\cdot)$, not how to design $U(\cdot)$. **Moreover, a data valuation method based on a fair utility function from stage 1 can still be prone to the shell company attack in stage 2, if it adopts the popular GSV approach there.**
>
> > **5. Clarification on experiment setup.**
> You commented: “Both the experimental setups and the results are difficult to follow...” and “The authors mention the test set for just one experiment…”
>
> Our experiments are designed to demonstrate specific contributions of our work: (i) identifying the shell company attack, (ii) developing a fast approximation algorithm for FGSV, and (iii) the robustness of our FGSV against shell company attack in two applications. Below, we clarify the objective and setup of each experiment:
>
> - **Sec 1 (B.1): Illustrating the shell company attack.** We used a supervised learning example to motivate it. The utility function $U(S)$ is the accuracy of a model trained on $S$ evaluated on a held-out test set. Thus, this setup involves data splitting, as described in line 714.
>
> - **Sec 4.1 (B.2): Benchmarking the approximation algorithm.** This experiment evaluates how efficiently and accurately Algorithm 1 approximates FGSV. Following [1], we used the Sum-of-Unanimity (SOU) game, where utility $U(S)$ is defined purely on index set $S$ and the exact Shapley value is analytically tractable. This is not a machine learning task and does not involve any dataset, therefore, test data, data splitting, or parameter $B$ are not applicable here. We set $\bar{s} = 10$, and partition the $n$ players evenly into four groups of size $n/4$.
>
> - **Sec 4.2 (B.3): First application to copyright attribution.** This experiment evaluates FGSV’s robustness against the shell company attack in the context of copyright sharing for generative AI. The utility is defined as the log-likelihood of a given newly generated image after fine-tuning with subset $S$, so no held-out test set is needed in this setup.
>
> - **Sec 4.3 (B.4): Second application to explainable AI.** The experiment evaluates whether FGSV maintains faithful data valuation in explainable AI. The goal is not to design a fair performance metric $U(S)$ as in ML fairness but to demonstrate the robustness against shell company attack in stage 2 for any given $U(S)$. This is a supervised learning setup, and the utility is defined as the test MSE. The splits of train/test data are described in line 859.
>
> We will make these distinctions more explicitly in the revised version.
>
> > **6. Related literature:**
> i). Existing works on handling copier attack [2,3,4]
> ii). Why not use Banzhaf instead of Shapley as the base method?
>
> Thank you for the references. Let’s discuss them.
>
> [2] proposes a specific form of utility function using Pearl’s “do” operation to differentiate between authentic contributors and copiers. Assuming access to the do-conditional probabilities, this paper essentially designed a utility function that rewards nothing to copiers. **Therefore, it can be naturally incorporated into our framework – we can simply use their utility function when we know the do-conditional probabilities and conveniently defend against the copier attack.**
>
> [4] establishes that by using a submodular utility function, Banzhaf [3] and leave-one-out can automatically defend against the following type of copier attack: a genuine contributor duplicates itself to gain higher total valuation, while Shapley does not.
>
> However, **this [4]'s approach is prone to "pure infringers" (copy attackers not associated with any genuine contributors)**, who enter the game, directly copying from valuable data points, without contributing anything in the first place.
>
> Specifically, on why we use Shapley not Banzhaf:
>
> 1. Our main target application is royalty sharing (or called “revenue allocation”). Therefore, additivity ("efficiency axiom") is an indispensable requirement. Banzhaf does not satisfy this requirement and cannot serve our target real-world application.
>
> 2. As aforementioned, Banzhaf’s defense against the copier attack still has a significant loophole.
>
> ---
> _**Reference**_
>
> [1] Li, W. and Yu, Y. One sample fits all: Approximating all probabilistic values simultaneously
> and efficiently. arXiv preprint arXiv:2410.23808, 2024.
>
> [2] Falconer, T., Kazempour, J. and Pinson, P., 2023. Towards replication-robust data markets. arXiv preprint arXiv:2310.06000.
>
> [3] Wang, J.T. and Jia, R., 2023, April. Data banzhaf: A robust data valuation framework for machine learning. In International Conference on Artificial Intelligence and Statistics (pp. 6388-6421). PMLR.
>
> [4] Han, D., Wooldridge, M., Rogers, A., Ohrimenko, O. and Tschiatschek, S., 2022. Replication robust payoff allocation in submodular cooperative games. IEEE Transactions on Artificial Intelligence, 4(5), pp.1114-1128

---

> > ### Comment · Reviewer_ct4R · 2025-08-04
> >
> > Thank you for the clarifications and detailed responses. I will raise my score with the expectation that the authors implement the proposed revisions based on all the reviewers' comments and the authors' rebuttals, such as fixing broken links, clarifying key distinctions, strengthening the literature review, and including a section on limitations and future works.

---

> > > ### Author Response · Authors · 2025-08-04
> > >
> > > Thank you very much!  We will revise the paper accordingly to incorporate the suggested improvements.

---

### Official Review · Reviewer_A8AR · 2025-07-03

**Clarity:** 4
**Significance:** 4
**Originality:** 4
**Rating:** 5
**Confidence:** 3

**Summary:**

This paper examines the usage of group level data valuation based on Shapley value.
The author makes this observation:  a malicious agent can game a mechanism that treats the groups as atomic entities,  using the so-called *shell company attack*, in which the malicious agent strategically splits its data between several shell companies.

The paper first result is that, to be robust to shell company attacks and satisfy the classical Shapley value axioms -- where anonymity has been adapted to the group setting--  one has no choice but to value a group of point as the sum of each point Shapley value.

This motivates the introduction of a Shapley value algorithm exploiting the group structure, in particular,
one can group the $U(S)$ terms in the Shapley value computation based on the size of |S| and the size of the  intersection  of $S$ with the group of interest (the one we want to compute the Shapley value).
This observation initiates a series of tricks that ends up with a novel, efficient algorithm.

Numerical experiments demonstrate competitiveness against sota as well as a scenario of shell company attack.

**Questions:**

A: How do you think theorem 3 presentation could be improved?
B: Do you have arguments to weights the pros and cons between coping against copier attack and shell company attack, are there other aspects we should have in mind?
C: How did you came up with condition 3? What does the literature tells us about it that made you believe it was the right condition to build the argument?

**Ethical Concerns:**

["NO or VERY MINOR ethics concerns only"]

**Final Justification:**

I keep my score, I believe this is a good paper and the authors answers my question in a satisfying manners.

**Limitations:**

yes

**Paper Formatting Concerns:**

nope

**Quality:**

4

**Strengths And Weaknesses:**

# Strengths
* The paper raises an important question by introducing the notion of shell company attack.
* This is a paper articulating several original observations about a very active field of research. Leading to a well motivated algorithm (+++). I find this paper very innovative.
* The reading of the paper is eased by the fact that the important ideas are well emphasized ("key observations..., pictures..").


# Weaknesses
* I find the current version of Theorem 3 hard to parse. I believe this should be improved.
* The paper mentions the copier attack only at the end (and they recognize the limitation of this pape with respect of this attack). In my first reading of the paper, I almost stop reading because they did not acknowledge it earlier (and this attack is what came to my mind immediately while reading). It would be good to mention the copier attack a bit ealier.
* (Still on the copier attack) The remark at the end falls like a clifthanger, and I think it is quite easy to find counter-measure against the authors's  defensive proposal (the "remove the copies" argument). I think this part should be elaborated a bit more.

---

> ### Author Rebuttal · Authors · 2025-07-31
>
> Thank you for your thoughtful and constructive comments.
>
> > **1. W1 & Question A: Clarification and Revision of Theorem 3**
>
> Thank you for this comment. We will revise Theorem 3 for readability. Here is the revised version.
>
> **Theorem 3.** _Algorithm 1 guarantees that for any truncation threshold $\bar s$ and sample sizes $m_1, m_2$, with probability at least $1-\delta$,_
>
> $$
> \left|\widehat{\mathrm{FGSV}}(S_0)-\mathrm{FGSV}(S_0)\right| \lesssim \bar s\sqrt{\frac{\log(n/\delta)}{m_1}}  +  \alpha_0(1-\alpha_0)\sqrt{\frac{\log(n/\delta)}{m_2}} \sum_{s=\bar s}^{n-1}\beta(s) + \alpha_0(1-\alpha_0)\bar s^{-\upsilon}.
> $$
> _Specifically, by choosing_
> $$
> \bar s  \asymp  \epsilon^{-1/\upsilon}, \quad m_1  \asymp  \epsilon^{-\frac{4+2\upsilon}{\upsilon}}  \log\bigl(n/\delta\bigr), \quad m_2  \asymp  \max\Bigl\\{1,  \epsilon^{-2}(\alpha_0(1-\alpha_0))^2(\log(n/\delta))^3\Bigr\\},
> $$
> _one obtains an $(\epsilon, \delta)$‑approximation of $\mathrm{FGSV}(S_0)$ with $O\bigl(n\cdot\mathrm{polylog}(n)\bigr)$ total utility evaluations._
>
> **Remark.** For the error bound, the first term comes from the Monte Carlo approximation error for small $s$, the second term comes from the Monte Carlo approximation error for large $s$, and the last term is the tail bound induced by the approximation in Theorem 2.
>
> > **2. W2, W3, & Question B: How can the proposed method (be extended to) handle the copier attack?**
>
> Thank you.  We will revise the paper and add more discussion on the copier attack, both earlier in the paper and in the discussion section.  NeurIPS does not allow revision during rebuttal, so let us give you an overview of the expanded discussion.
>
> The copier attack impacts both group and individual valuation methods.  The comments from you and other reviewers pointed out two remedies:
> - [1] suggests designing the utility function using Pearl’s “do operation”.  **This defense can be incorporated into our framework, and let our method also defend against the copier attack.**
> - [2] suggests using replication-robust data valuation measures, such as Banzhaf or leave-one-out, to replace Shapley.  This approach can automatically fend off one subtype of copier attack by discouraging existing contributors from self-replication, **but it does not defend against “pure infringers”** who copy valuable players without contributing any original content.
>
> Overall, it remains an **open challenge** to defend against the copier attack for **general** utility functions, while keeping the defense against the shell company attack up.  In view of our Theorem 1, a satisfactory solution will be based on our FGSV.  Therefore, this paper **pioneers an important first step** in the exploration of safe group valuation methods.
>
> > **3. Question C: Motivation of Condition (3) in Proposition 1**
>
> Prudence is a classical concept in economics: a consumer is called “prudent” if they save more when facing greater future income risk – mathematically, this is characterized as the utility’s third derivative being nonnegative [5].  In machine learning applications, several popular utility function curvatures satisfy the prudence condition, such as
> - Power‑law utilities ($\bar U(s)\propto s^a$ with $0<a<1$)
> - Logarithmic utilities ($\bar U(s)\propto \log s$)
>
> For instance, empirical studies report that scaling laws in deep learning [3] and large‑language‑model performance [4] follow power‑law trends in sample size \(s\), thereby also satisfying the prudence condition.
>
> ---
> _**References**_
>
> [1] Falconer, T., Kazempour, J. and Pinson, P., 2023. Towards replication-robust data markets. arXiv preprint arXiv:2310.06000.
>
> [2] Han, D., Wooldridge, M., Rogers, A., Ohrimenko, O. and Tschiatschek, S., 2022. Replication robust payoff allocation in submodular cooperative games. IEEE Transactions on Artificial Intelligence, 4(5), pp.1114-1128
>
> [3] Hestness, Joel, Sharan Narang, Newsha Ardalani, Gregory Diamos, Heewoo Jun, Hassan Kianinejad, Md Mostofa Ali Patwary, Yang Yang, and Yanqi Zhou. 2017. “Deep Learning Scaling Is Predictable, Empirically.” arXiv [Cs.LG].
>
> [4] Kaplan, Jared, Sam McCandlish, Tom Henighan, Tom B. Brown, Benjamin Chess, Rewon Child, Scott Gray, Alec Radford, Jeffrey Wu, and Dario Amodei. 2020. “Scaling Laws for Neural Language Models.” arXiv [Cs.LG].
>
> [5] Sandmo, A. 1970. “The Effect of Uncertainty on Saving Decisions.” The Review of Economic Studies 37 (3): 353.

---

> > ### Comment · Reviewer_A8AR · 2025-08-04
> >
> > Thank you for your helpfull answers. I will keep my score.

---

> > > ### Author Response · Authors · 2025-08-04
> > >
> > > Thank you very much!  We really appreciate it.

---

### Official Review · Reviewer_V1yM · 2025-07-03

**Clarity:** 3
**Significance:** 3
**Originality:** 3
**Rating:** 4
**Confidence:** 4

**Summary:**

This article addresses the problem of dataset valuation, where the goal is to fairly assess the contributions of agents—represented by datasets—using Shapley values. Traditional state-of-the-art (SOTA) methods are vulnerable to *shell company attacks*, where an agent splits their dataset into multiple subsets to unfairly increase their total valuation.

To overcome this, the authors propose the Faithful Group Shapley Value (FGSV), which evaluates each dataset as the sum of the Shapley values of its individual data points. Intuitively, this fine-grained approach prevents agents from benefiting through artificial data splitting, as FGSV assigns value at the most granular level.

FGSV is shown to be the only dataset valuation method that satisfies a set of five axioms: the four standard Shapley axioms adapted to the group setting, plus *faithfulness*—the property that ensures robustness against shell company attacks.

Although computing FGSV exactly is computationally expensive, the authors develop a fast and accurate approximation algorithm that works under mild regularity conditions. They demonstrate scenarios where these conditions are met and validate the proposed method with extensive experiments, showing strong performance compared to various SOTA alternatives.

**Questions:**

1. How accurate is it to evaluate the importance of a dataset by aggregating pointwise valuations? Since the value function $u$ is not necessarily linear, the dataset-level Shapley value  $SV_{S_0}(u)$ can differ significantly from the sum $\sum_{i\in S_0} SV_i(u)$. Could you elaborate on the implications of this discrepancy?

2. How does your method perform when Assumptions 1 and 2 do not hold? Specifically, do the three empirical scenarios presented in the paper satisfy these assumptions? If not, how robust is the method in such cases?

3. Why is the area under the convergence curve an indicator of accuracy? While it reflects convergence speed, does it guarantee convergence to the correct values, particularly in light of the first concern regarding the dataset vs. pointwise valuation?

**Ethical Concerns:**

["NO or VERY MINOR ethics concerns only"]

**Final Justification:**

The article is good. However, I have decided to keep my score due to the insufficient literature review. The authors argue that shell company attacks are their main novelty, but the idea of coalitions splitting into smaller coalitions is quite standard in cooperative game theory. Moreover, the claim that the only way to prevent such attacks is to reward datasets by the sum of the Shapley values of their individual data points feels somewhat disappointingly intuitive.

**Limitations:**

The article does not adequately address the limitations of the proposed method. In particular, the concerns raised in Questions 1 and 2 are critical and deserve a thorough discussion.

**Quality:**

3

**Strengths And Weaknesses:**

Strengths
- The article is clearly written and easy to follow.
- The addressed problem is both interesting and novel, particularly the focus on robustness against shell company attacks.
- The paper is mathematically rigorous, with well-justified theoretical assumptions.
- The proposed method is validated through a comprehensive set of experiments.

Weaknesses
– The method evaluates the importance of a dataset as the sum of the valuations of its individual data points, which may fail to capture the overall contribution of the dataset as a unit.
- The performance of the proposed method under violations of the regularity conditions is not thoroughly investigated.
- The literature review is insufficient, making it difficult to fully assess the contribution relative to existing work.
- The paper lacks a dedicated discussion of limitations, especially regarding scalability and applicability in less ideal scenarios.

---

> ### Author Rebuttal · Authors · 2025-07-31
>
> Thank you for your valuable comments.
>
> > **1. W1 & Q1: FGSV is defined as the sum of individual valuations.  This may fail to capture the overall contribution of the dataset, since the valuation function $u$ is nonlinear. Elaborate on the implications of the discrepancy between the dataset-level Shapley value $\mathrm{GSV}(S_0)$ and the sum $\mathrm{FGSV}(S_0)=\sum_{i\in S_0}\mathrm{SV}(i)$.**
>
> Allow us to clarify a potential conflation: our FGSV adds up individual Shapley values (SV), not individual utility functions $u$.  Although both SV and utility can be interpreted as “valuation function” in some sense, they are different concepts.  To illustrate, let’s consider the following metaphor:
>
> - Utility $U(S)$ is the total revenue earned by the joint work of group $S$.
> - $\mathrm{SV}(S)$ is the total salary allocated to $S$.
>
> For an individual $i$, define $U(i)$ and $\mathrm{SV}(i)$ by setting $S=\\{i\\}$. Note that $U(i)$ is the solo revenue, but $\rm{SV}(i)$ is not simply $U(i)$; it is computed competitively to capture the uniqueness and indispensability of $i$’s contribution – see Eqn. (1).
>
> We agree that summing up $U(i)$ ignores the extra merit of joint force. However, what we propose is that group salary be the sum of its members’ individual salaries, which already encodes interaction effects across possible coalitions. This is both intuitively sensible and, in view of our Theorem 1, mathematically necessary.
>
> In contrast, $\mathrm{GSV}(S_0)$ treats $S_0$ as a whole and fails to account for its members' unique contributions, which makes it vulnerable to group manipulations like the shell company attack.
>
>
> > **2. W2, W4 & Q2: Discussion of limitation/less idea scenarios: what if Assumptions 1 and 2, and regularity conditions are violated?  Are they satisfied in the three data examples?**
>
> Importantly, we clarify that there are two types of assumptions. The first type is about the data-generating process, over which a user has little control (e.g., assuming data follows a normal distribution and building a method upon it). In this case, it is crucial to assess the impact of assumption violation by sensitivity analyses.
>
> We emphasize that in this paper, we make no assumptions in the first type.
>
> **Instead, our Assumptions 1 and 2 and regularity conditions belong to the second category, that is, assumptions on things that the user has control over.**  Notice that all these assumptions eventually regard the utility function, and the practitioner (user) chooses the utility function. For example, many machine learning problems essentially predict response $y$ based on predictor $x$. Then utility depends on: 1) the prediction method $f: x\to y$, as well as the optimization algorithm; and 2) the performance measure, such as $\|\hat y-y\|$ under a proper norm. The user can choose both 1) and 2), and that fully determines the utility.
>
> **How common is it that a utility function satisfies our assumption?  Our understanding is “very often”.** In our three data examples, Assumption 1 is satisfied by our design of the performance measure. For example, we use the log-likelihood on a new image (normalized to a compact set) in the copyright attribution task, and a bounded mean squared error for disease progression in the explainable AI example (as the maximum follow-up time is bounded). Assumption 2 is satisfied under the standard algorithm and model choice, such as using a DNN with smooth activation functions and bounded parameters trained with SGD. In our experiments, we used advanced optimizers like Adam beyond standard SGD. Our stable results demonstrate that our method performs robustly well even when these conditions are not strictly met.
>
> **Our assumptions are quite standard compared to other works [2, 3, 4].** One can certainly think of a scenario that violates our assumptions, where the utility is built based on an unpatterned prediction method with rather arbitrary behaviors, then to our best knowledge, no other existing theoretical results in this area will apply, either.
>
>
> > **3. W3: Literature review seems insufficient, making it difficult to fully assess the contribution relative to existing work.**
>
> Generic literature: we cited prior works on data valuation, including non-Shapley-type methods and other Shapley-type methods.  This part is similar to other papers on Shapley value.
>
> Most related literature: in fact, the topic “group data valuation” has emerged very recently.  We cited major advances, as bib entries [6,14,32,35] in our paper.
>
> **Our contribution’s significance:** the group-as-individual (GaI) paradigm dominates existing literature.  However, in this paper, we proposed a new attack, i.e., the shell company attack, that exposes a fundamental flaw in GaI.  We proposed a new method FGSV and proved that it uniquely defends against this new attack.  Moreover, we devised a fast algorithm to numerically approximate FGSV.
>
> > **4. W4: Method scalability?**
>
> Estimating Shapley values involves evaluating the utility function on many subsets of the data, which can be computationally expensive. However, this expense is unavoidable to any Shapley‐based estimator.  Crucially, our FGSV algorithm provably **reduces** the number of utility evaluations needed for an $(\epsilon,\delta)$‑approximation: compared to prior methods, it requires far fewer calls to the utility oracle while still delivering the same accuracy guarantees.
>
> > **5. Q3: Why use AUCC to measure performance (Section 4.1)?  AUCC doesn’t show methods’ accuracy at convergence.**
>
> Our adoption of AUCC follows [1], who used it to compare the convergence behaviors of several individual Shapley value estimators. AUCC summarizes how quickly and stably an estimator approaches the ground truth by integrating the error across the intermediate phase.
>
> We agree that AUCC does not show the “final” accuracy of an estimator. To address this, we also report the **absolute relative error (ARE)** of the final estimates:
> $$
> \mathrm{ARE}(S_k)=\left|\frac{\mathrm{FGSV}(S_k)-\widehat{\mathrm{FGSV}}^{(20000)}(S_k)}{\mathrm{FGSV}(S_k)}\right|.
> $$
> The table below presents ARE (mean and sd across 30 runs) for the first group $S_1$. Our method consistently achieves the lowest ARE. Results for other groups are similar and omitted here for brevity.
> ***
> ||Permutation|Group Testing|Complement|One-for-All|KernelSHAP|Unbiased KernelSHAP|LeverageSHAP|**FGSV**|
> |:---|:---:|:---:|:---:|:---:|:---:|:---:|:---:|:---:|
> |$n=64$|0.04945 (0.03597)|0.18083 (0.11655)|0.05496 (0.11161)|0.00819 (0.00674)|0.08503 (0.05977)|0.02690 (0.02030)|0.07921 (0.05467)|**0.00161 (0.00047)**|
> |$n=128$|0.07252 (0.04468)|0.24966 (0.16843)|0.19794 (0.05634)|0.01531 (0.04132)|0.15551 (0.09972)|0.01930 (0.01848)|0.08117 (0.06099)|**0.00139 (0.00056)**|
> |$n=256$|0.08818 (0.06402)|0.37767 (0.26249)|0.83911 (0.04193)|0.05433 (0.01457)|0.26393 (0.20925)|0.02536 (0.01830)|0.13617 (0.10424)|**0.00100 (0.00075)**|
>
> ***
> _**References**_
>
> [1] Li, W. and Yu, Y. One sample fits all: Approximating all probabilistic values simultaneously
> and efficiently. arXiv preprint arXiv:2410.23808, 2024.
>
> [2] Jia, R., et al. 2019. “Towards Efficient Data Valuation Based on the Shapley Value.” Edited by Kamalika Chaudhuri and Masashi Sugiyama. International Conference on Artificial Intelligence and Statistics abs/1902.10275 (February): 1167–76.
>
> [3] Jia, R., et al. 2019. “Efficient Task-Specific Data Valuation for Nearest Neighbor Algorithms.” Proceedings of the VLDB Endowment International Conference on Very Large Data Bases 12 (11): 1610–23.
>
> [4] Ghorbani, A., et al. 2020. “A Distributional Framework for Data Valuation.” International Conference on Machine Learning abs/2002.12334 (February): 3535–44.

---

> > ### Comment · Reviewer_V1yM · 2025-08-01
> >
> > Thank you very much for your responses. All my points have been clarified.
> > Still, please keep in mind that, as also observed by other reviewers, the article would benefit from a more comprehensive literature review (e.g., including related work on shell company attacks).

---

> > > ### Author Response · Authors · 2025-08-01
> > >
> > > Thank you for your reply.
> > >
> > > We will certainly follow your advise and keep the comprehensiveness of literature in mind in the next revision.  Here is a list for your further reference on what specific aspects we plan to add to literature review:
> > > * Bullet point 1 in our reply to Reviewer gZ5H
> > > * Bullet point 2 in our reply to Reviewer A8AR
> > > * Bullet point 6 in our reply to Reviewer ct4R
> > > * Bullet point 5 in our reply to Reviewer R2x5
> > >
> > > We would like to clarify that the shell company attack is our original contribution — we do not believe there is existing literature on this.
> > >
> > > Thank you again.

---

### Official Review · Reviewer_gZ5H · 2025-07-22

**Clarity:** 3
**Significance:** 2
**Originality:** 2
**Rating:** 4
**Confidence:** 3

**Summary:**

The authors address the problem of dataset valuation based on Shapley values (called group-level Shapley), where the objective is to derive Shapley semi-values for each dataset participating into a machine learning task. This differs from classical datum valuation, where one player is a data sample from a fixed dataset. Group-level valuation is subjected to a specific adversarial attack (shell attack), where other players can split their datasets to change the value of the current player. To cope with that issue, the authors are proposing a generalisation of the Shapley value for dataset valuation by introducing an additional axiom introducing a fairness constraint to address shell attacks. They provide an algorithm computationally efficient and illustrate the benefits of the latter on several experiments.

**Questions:**

See above.

**Ethical Concerns:**

["NO or VERY MINOR ethics concerns only"]

**Quality:**

3

**Strengths And Weaknesses:**

**Strengths**
* The Faithful Group Shapley Value proposed by the authors is, up to my knowledge, novel.
* The proposed algorithm is original, scalable (from $O(n^2)$ to $O(n)$ computational cost up to polylogarithmic terms, where $n$ is the number of players). Note that $O(n)$ is the SOTA computational cost in terms of number of utility evaluations.
* All claims are theoretically justified, albeit I did not check the full supplementary material.
* Experiments are revelant and indeed show the benefits of the proposed approach.

** Weaknesses**
* Could the authors provide more insights on why they choose to focus on the shell attack compared to others? A dedicated paragraph on the most impactful and common attacks would be valuable to know at which extent this novel methodology will be relevant for the community.
* The authors are introducing FGSV with a new faithful axiom but are considering an algorithm approximating the latter and hence potentially loosing the faithful axiom and the ability to cope with shell attacks. Could the authors comment on that point?

---

> ### Author Rebuttal · Authors · 2025-07-31
>
> Thank you for your valuable comments.
>
> > **1. Why choose to study the shell company attack?  What are other common attack methods?**
>
> **The shell company attack is our original contribution.**  We identified a significant vulnerability in popular group data valuation methods, which has not been discovered in the existing literature.
>
> The copier attack is the most significant other type of attack that we know, which has been addressed in the data valuation literature [1].  Shell company and copier are completely different attacks: copier attack duplicates existing data; while shell company strategically manipulates data grouping.
>
> Overall, the topic of “group data valuation” has only emerged very recently [2,3].  Works on attack and defense of data valuation exclusively concentrate on individual data valuation.  **Our paper is the first, pioneering work that addressed the unexplored topic of attack and defense for group data valuation.**
>
> Thanks to your comments, we will revise the presentation of our paper’s contributions and make the above points clearer in the next version.
>
> > **2. Approximately computing FGSV may break axiom compliance and lose defense against the shell company attack**
>
> It is necessary and a common practice to compute numerical approximations to Shapley values in the literature, since the exact Shapley values are computationally prohibitive.
>
> Our Theorem 3 guarantees that the output of the algorithm approximately obeys the axioms, including the faithfulness axiom.  Numerical results such as Figures 1 and 3 clearly demonstrate our method’s robustness against the shell company attack.
>
> ***
> _**References**_
>
> [1] Falconer, T., Kazempour, J. and Pinson, P., 2023. Towards replication-robust data markets. arXiv preprint arXiv:2310.06000.
>
> [2] J. T. Wang, Z. Deng, H. Chiba-Okabe, B. Barak, and W. J. Su. An economic solution to copyright challenges of generative AI. arXiv preprint arXiv:2404.13964, 2024.
>
> [3] J. Wang, Y. Chen, and P. Giudici. Group Shapley with robust significance testing and its application to bond recovery rate prediction. arXiv preprint arXiv:2501.03041, 2025.

---

### Note · Authors · 2025-08-14

We thank all reviewers for their comments and discussion.  Here, we briefly recap our paper and key discussion points in the rebuttal.

**Background:**
* Proper data pricing is foundational for a healthy, sustainable data market.
* While most prior work regards individual data valuation, real royalty sharing requires group data valuation.
* This line of work has only started very recently.  The predominant approach is Group Shapley Value (GSV).

**Contribution highlights:**
1. We propose **shell company attack**, a novel attack that exposes a fundamental vulnerability of GSV.
2. We propose **Faithful Group Shapley Value (FGSV)** that can fence off this attack.
3. We prove that FGSV is the only defence method for general utility.
4. We discovered original mathematical insights that enable fast computation of FGSV.

**Reviewers commonly acknowledged the following strengths:**
1. Addressing an **important and novel** problem.
2. **Novelty and scalability** of the proposed method and **fast algorithm** based on **original mathematical insights**.
3. Thorough validation through **comprehensive experiments**.

**Rebuttal gist:**
* Common question: the copier attack?
    * We can use a utility that resists this attack (Falconer et al, 2023).
    * Some reviewers suggested replacing Shapley by other semi-values (Han et al, 2022), but this defense is prone to “pure infringers”.
    * Defending against the copier attack for general utility remains an open problem.
* Reviewer V1yM: robustness to assumption violations.
    * Our assumptions solely regard the utility and are verifiable in practice.
    * Users can choose “well-behaved” utilities that satisfy them; we provide examples showing that commonly used utilities typically do.
* Reviewer ct4R: relation to machine learning fairness.
    * Data valuation has two stages: (1) design a utility function, where fairness considerations reside; (2) aggregate utility values into valuations, which is our paper’s focus. Our FGSV is compatible with ML fairness by pairing it with a fairness-aware utility function, though fairness design itself is outside the scope of this paper.
* Other comments asked for clarifications on technical details and minor fixes (typos, broken links, etc).  We specified the changes to include in the revision to incorporate these constructive suggestions.

We believe the proposed method offers both theoretical novelty and practical value, addressing a novel attack in group data valuation.

---

### Decision · Program_Chairs · 2025-09-17

**Decision:**

Accept (poster)

**Comment:**

This paper considers the problem of dataset valuation using the Shapley value, and the objective is to be "robust" against other players that could split/merge their dataset to lower the value of a single player.

This idea is quite new and interesting, and of practical interest.

This is the reason why, even if the reviewers had some questions and concerns, they were all rather positive about this paper. So I went through it and it is, indeed, quite interesting.

For this reason, I am also in favor of accepting this paper to NeurIPS 2025 !